# Highest fusion performance without harmful edge energy bursts in tokamak

S. K. Kim[1,10], R. Shousha[1,10], S. M. Yang [1,10], Q. Hu[1], S. H. Hahn[2], A. Jalalvand[3], J.-K. Park[4], N. C. Logan[5], A. O. Nelson[5], Y.-S. Na[4], R. Nazikian[6], R. Wilcox[7], R. Hong[8], T. Rhodes[8], C. Paz-Soldan[5], Y. M. Jeon[2], M. W. Kim[2], W. H. Ko[2], J. H. Lee[2], A. Battey[5], G. Yu[9], A. Bortolon[1], J. Snipes[1] & E. Kolemen[1,3] ✉

The path of tokamak fusion and International thermonuclear experimental reactor (ITER) is maintaining high-performance plasma to produce sufficient fusion power. This effort is hindered by the transient energy burst arising from the instabilities at the boundary of plasmas. Conventional 3D magnetic perturbations used to suppress these instabilities often degrade fusion performance and increase the risk of other instabilities. This study presents an innovative 3D field optimization approach that leverages machine learning and real-time adaptability to overcome these challenges. Implemented in the DIII-D and KSTAR tokamaks, this method has consistently achieved reactor-relevant core confinement and the highest fusion performance without triggering damaging bursts. This is enabled by advances in the physics understanding of self-organized transport in the plasma edge and machine learning techniques to optimize the 3D field spectrum. The success of automated, real-time adaptive control of such complex systems paves the way for maximizing fusion efficiency in ITER and beyond while minimizing damage to device components.

For a fusion energy source to be economically competitive in the global energy market, it must achieve a high fusion triple product ($n\tau T$)[1] with sufficient plasma density ($n$), temperature ($T$), and energy confinement time ($\tau$) while sustaining fusion reactions. In other words, the fusion plasma requires a sufficient figure of merit ($G \propto n\tau T$)[1–6] for high fusion performance, which increases with plasma confinement quality ($H_{89}$)[7], where $H_{89}$ is normalized energy confinement time. For example, the International thermonuclear experimental reactor (ITER) requires $G > 0.4$ and $H_{89} > 2$ to achieve[1] its objective (a fusion power ten times higher than the input heating power). One of the leading approaches[8] towards this goal is a tokamak operated robustly in the high confinement mode (H-mode)[9], characterized by a narrow edge transport barrier (or confinement pedestal) responsible for

significantly elevated plasma pressures within the device. This pedestal has demonstrated notable benefits by enhancing $G$, thereby improving the fusion economy. However, the H-mode has a high-pressure gradient at the edge (pedestal), which introduces significant risks to reactor operation, mainly due to emerging dangerous edge energy bursts as a result of a plasma instability known as edge localized modes (ELMs)[10]. These edge bursts cause rapid relaxations in pedestal plasma energy, leading to intense transient heat fluxes on reactor walls, resulting in undesirable material erosion and surface melting. The predicted heat energy reaches ~20 MJ/m², unacceptable in a fusion reactor[11,12]. Consequently, for tokamak designs to become a viable option for fusion reactors, reliable methods must be developed to routinely suppress edge burst events without affecting $G$.

[1]Princeton Plasma Physics Laboratory, Princeton, NJ, USA. [2]Korea Institute of Fusion Energy, Daejeon, South Korea. [3]Princeton University, Princeton, NJ, USA. [4]Seoul National University, Seoul, South Korea. [5]Columbia University, New York, NY, USA. [6]General Atomics, San Diego, CA, USA. [7]Oak Ridge National Laboratory, Oak Ridge, TN, USA. [8]University of California Los Angeles, Los Angeles, CA, USA. [9]University of California Davis, Davis, CA, USA. [10]These authors contributed equally: S. K. Kim, R. Shousha, S. M. Yang. ✉e-mail: ekolemen@princeton.edu

Numerous endeavors have been made to mitigate edge burst events through various approaches. These include exploring scenarios such as small[13–18] or non-ELMing[19–24] regimes, wherein edge bursts are spontaneously reduced or dissipated. Another effective approach is to utilize resonant magnetic perturbations (RMPs) by external 3D field coils[25–28], which have proven to be one of the most promising methods for edge burst suppression. The typical external coils surrounding the plasma to generate 3D fields are shown in Fig. 1. By reducing the pedestal[29–39], 3D fields effectively stabilize energy burst in the edge region[40]. This stabilizing effect offers a significant advantage by actively minimizing the bursty heat flux and seed perturbation that can trigger core instability. Therefore, the ITER baseline scenario relies on 3D-field to achieve an edge-burst-free burning plasma in a tokamak for the first time.

However, this scenario comes at a significant cost, resulting in a significant deterioration of $H_{89}$ and $G$ compared to standard high-confinement plasma regimes, thus depleting economic prospects. Moreover, the 3D field also raises the risk of disastrous core instability, known as a disruption, which is even more severe than an edge burst. Thus, the safe accessibility and compatibility of edge-burst-free operation with high confinement operation requires urgent exploration.

This work reports on an innovative and integrated 3D-field optimization on both KSTAR and DIII-D tokamaks for the first time by combining machine learning (ML), adaptive[41,42], and multi-machine capabilities for automatically accessing and achieving an almost fully edge-burst-free state while boosting the plasma fusion performance from its initial burst-suppressed state, which is a significant milestone toward edge-burst-free operation for future reactors. This is accomplished by real-time exploitation of hysteresis between edge-burst-free onset and loss to enhance plasma confinement while extending the ML capability in capturing physics and optimizing nuclear fusion technology[43–46].

This integration facilitates (1) highly enhanced plasma confinement, reaching the highest fusion $G$ (see Fig. 2) among ELM-free scenarios in two machines with an increase in $G$ up to 90%, (2) fully automated 3D-field optimization for the first time by using an ML-based 3D-field simulator, and (3) concurrent establishment of burst suppression from the very beginning of the plasma operation, achieving nearly complete edge-burst-free operation close to the ITER-relevant level. Such an achievement paves a vital step for future devices such as ITER, where relying on empirical RMP optimization is no longer a viable or acceptable approach.

This paper is organized as follows. We first explain the integrated 3D-field optimization algorithms. The contribution of ML and adaptive schemes in the optimization process is introduced in the following sections. Then, the utilization of early 3D (or RMP) algorithms toward a complete ELM-free operation and underlying physics phenomena allowing the burst-free operation with high $G$ are presented. Lastly, the discussions on the application in ITER and future reactors are drawn in summary.

## Results

### Fully automated optimization of 3D-field using ML-surrogate model

The key to stable and robust ELM suppression is maintaining a sufficient edge 3D field ($B_{edge}$) for the ELM suppression while minimizing the core perturbation ($B_{core}$) or $r_B = B_{core}/B_{edge}$, which can induce seed perturbations and radial transport in the core, making the onset of plasma disruptions easier. $B_{edge}$ and $r_B$ can be controlled by adjusting the RMP current (or amplitude, $I_{RMP}$) and current distribution among external 3D coils (e.g., 3D waveform). For these reasons, in the present experiments, a series of discharges are used to find an optimized 3D waveform for safe ELM suppression. The successful ELM suppression in the previous studies also relies on the empirically derived 3D setup. However, this trial-and-error approach isn't viable in a fusion reactor, where an unmitigated high current disruption can greatly reduce the machine's lifespan[47]. Achieving reliable ELM suppression in a reactor requires a first-principle strategy to determine the 3D waveform adaptively.

In this context, this work introduces the ML technique to develop the novel path of automated 3D coil optimization and demonstrate the concept for the first time. This approach exploits the physics-based optimization scheme of 3D waveform[48] based on plasma equilibrium and ideal 3D response from GPEC simulation[49]. This method has been validated across multiple devices[48,50,51] and extensively tested on KSTAR, which has flexible 3D coils with three rows, resembling ITER's configuration. This approach effectively predicts the optimal 3D coil setup that minimizes $r_B$ to ensure safe ELM suppression. However, its computational time, taking tens of seconds, hinders real-time applicability, limiting its use to pre-programmed or feed-forward strategies.

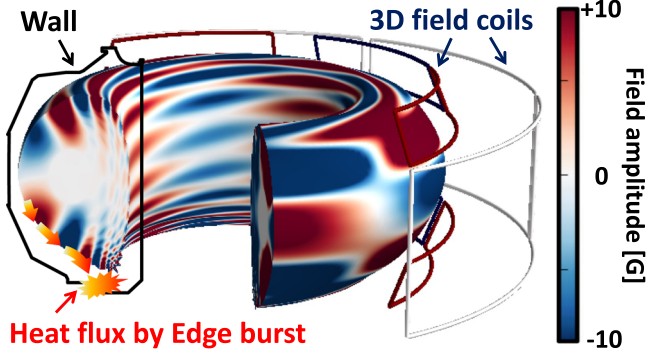

**Fig. 1 | 3D-field coil structure in tokamak.** Schematic diagram of 3D field coil and edge energy-burst in DIII-D tokamak. Color contour shows typical 3D-field amplitude formed by coils.

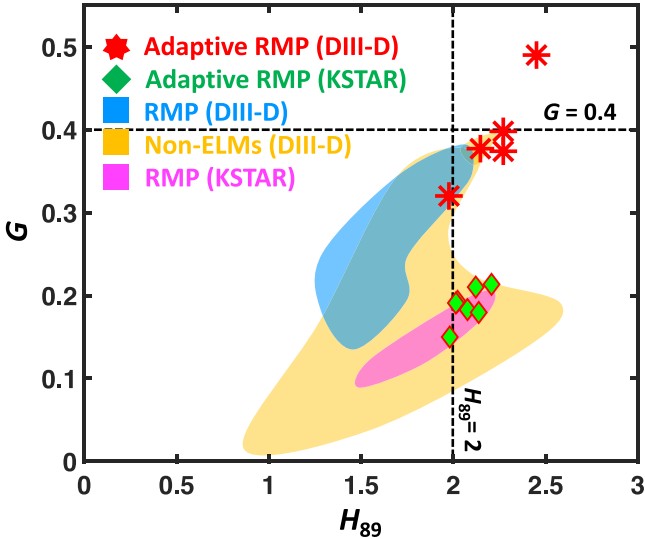

**Fig. 2 | Performance comparison of ELM-free discharges in DIII-D and KSTAR tokamaks.** Normalized energy confinement time ($H_{89}$) versus the figure of merit ($G$) at ELM-free state. These cover conventional (blue and purple regime), adaptive RMP, and various non-ELMing scenarios (orange regime), including QH[23,24], I-mode[21,22], and EDA-H mode[19,20] in DIII-D. The red star and green diamond markers show the adaptive RMP discharges in DIII-D and KSTAR, respectively. The dashed lines indicate the ITER-relevant level[1,62] required to achieve their baseline target parameters (normalized beta ($\beta_N$) = 1.8, edge safety factor ($q_{95}$) = 3, $H_{89}$ = 2). A detailed plot for this figure can be found in Supplementary Fig. 1.

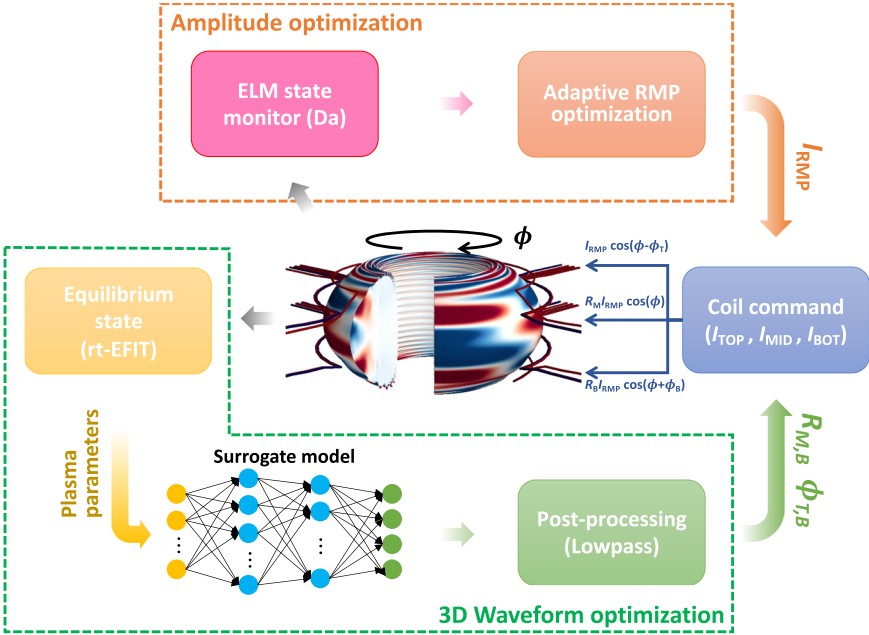

**Fig. 3 | Machine-learning-based real-time RMP optimization algorithm.** Schematic diagram of integrated RMP optimization scheme in KSTAR with ML-surrogate model (ML-3D) with 3D-coil variables ($I_{RMP}$, $I_{TOP}$, $I_{MID}$, $I_{BOT}$, $R_{M,B}$, and $\phi_{T,B}$).

To overcome such limitations, a surrogate model (ML-3D) of GPEC code has been developed to leverage the physics-based model in real time. This model uses ML algorithms to accelerate the calculation time to the ms scale, and it is integrated into the adaptive RMP optimizer in KSTAR. ML-3D consists of a fully connected multi-layer perceptron (MLP) which is driven by nine inputs, the total plasma current ($I_P$), edge safety factor ($q_{95}$), global poloidal beta ($\beta_P$), global internal inductance ($l_i$), the coordinates of X-points on the $R$–$Z$ plane ($R_X$, $Z_X$), and the plasma elongation ($\kappa$). These parameters are derived from real-time equilibrium[52] calculations and are normalized to have zero mean and unit variance per input feature overall training set. The outputs of the model are coil configuration parameters ($R_M$, $R_B$, $\phi_T$, $\phi_B$), which determine the relations between coil current distribution across the top ($I_{TOP}$), middle ($I_{MID}$), and bottom ($I_{BOT}$) 3D coils. Here, $R_M = I_{MID}/I_{TOP}$, $R_B = I_{BOT}/I_{TOP}$, and $\phi_{T,B}$ is the toroidal phasing of the top and bottom coil currents relative to the middle coil (see Fig. 3). In order to train this model, the GPEC simulations from 8490 KSTAR equilibria are utilized.

As shown in Fig. 3, the algorithm adaptively changes $I_{RMP}$ in real-time by monitoring the ELM state using the $D_\alpha$ signal. This maintains a sufficient edge 3D field to access and sustain the ELM suppression. At the same time, the 3D-field optimizer adjusts the current distribution across the 3D coils using the output of ML-3D, which guarantees a safe 3D field for disruption avoidance. This model generates the relations between coil currents ($R_{M,B}$, $\phi_{T,B}$) at every 1 ms given equilibrium state. Figure 4a–d illustrates the performance of ML-3D with a randomly selected test discharge, showing good agreement between the offline and ML-3D outputs. A low-pass filter is applied to prevent overly rapid changes in the 3D-coil commands that could result in damage to the coils.

In KSTAR discharge (#31873) with plasma current, $I_p = 0.51$ MA, edge magnetic pitch angle, $q_{95} \approx 5.1$, and ~2.5 MW of neutral beam injection heating, the ML-integrated adaptive RMP optimizer is triggered at 4.5 s and successfully achieves fully-automated ELM suppression without the need for pre-programmed waveforms. As shown in Fig. 5a, $I_{RMP}$ starts increasing at 4.5 s with a rate of 3 kA/s to access the ELM suppression, while $R_{M,B}$ and $\phi_{T,B}$ adjust simultaneously. As a 3D coil setup is automatically optimized by ML-3D during the entire discharge, safe ELM suppression is achieved at 6.2 s.

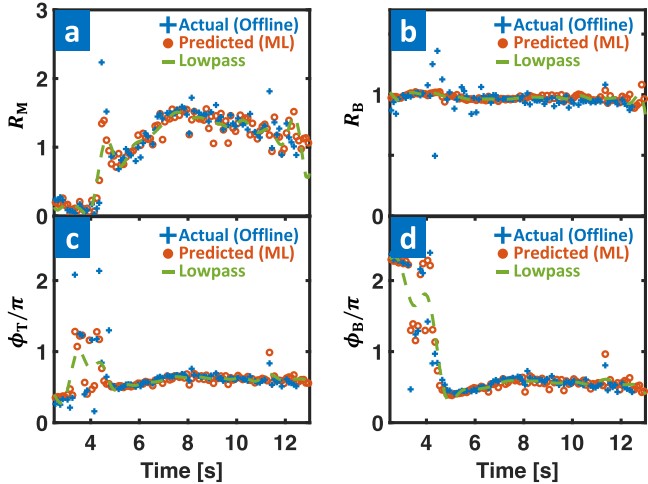

**Fig. 4 | ML-3D model performance. a–d** Validating comparison of the model using a test case, showing actual (blue), predicted (orange), and low-pass filtered (green) 3D-coil variables ($R_{M,B}$ and $\phi_{T,B}$) in time.

During the optimization process (see Fig. 5e), ML-3D maintains $r_B$ at a level similar to the empirically optimized 3D setup (standard, $R_{M,B} = 1$, $\phi_{T,B} = \pi/2$). Interestingly, ML-3D achieves such a favorable $r_B$ even with different coil configurations from the empirical (standard) case, highlighting the capability of ML-3D in finding the effective physics-informed path of 3D-field optimization. From the plot, it is apparent that ($\epsilon = r_{B,ML}/r_{B,STD}$) stays near or below unity, where $r_{B,ML}$ and $r_{B,STD}$ are $r_B$ from ML-3D and standard setup, respectively. Furthermore, ML-3D performs better than the empirical setup in the early stages of ELM suppression (<6 s), showing much lower $r_B$ than the standard case (or $\epsilon < 1$). This behavior is particularly beneficial as keeping $r_B$ small in the early ELM control phase is key to avoiding disruption, explaining how the successful automated ELM suppression is achieved. Therefore, these results show the 3D-ML as a viable solution for automated ELM-free access. Furthermore, unlike conventional RMP experiments, there is a rapid increase in $\beta_P$ (or confinement) after entering ELM suppression through the adaptive control of $I_{RMP}$. This

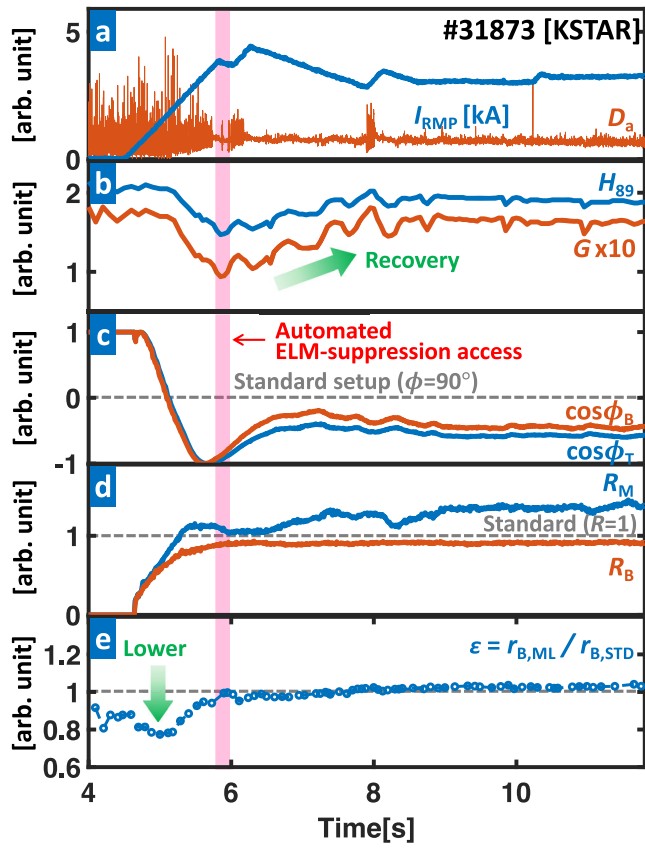

**Fig. 5 | Plasma parameters for a fully automated ELM suppression discharge (#31873) with integrated RMP optimization. a** RMP coil current ($I_{RMP}$, blue) and particle recycling light ($D_\alpha$ emission, orange) near outer divertor target. **b** Normalized confinement time ($H_{89}$, blue) and figure of merit ($G$, orange). **c** Phasing between top/middle ($\phi_T$, blue) and middle/bottom coils ($\phi_B$, orange). **d** Current amplitude ratio between of top/middle ($R_M$, blue) and top/bottom coils ($R_B$, orange). **e** Ratio ($\epsilon = r_{B,ML}/r_{B,STD}$) of 3D-coil induced $r_B$ from ML-3D ($r_{B,ML}$) and predicted one using an empirical configuration ($r_{B,STD}$). Smaller $\epsilon$ means lower $r_B$ than the one by standard (empirical) setup. The gray dotted line in **c**, **d** shows the 3D-coil configurations from a standard (empirical) setup. The red-colored area highlights the automated access to the ELM-suppressed state without pre-programmed 3D fields. The green arrows in **b**, **e** highlight the confinement recovery and enhanced field optimization by the control algorithm. The optimization algorithm is triggered at 4.5 s.

change can amplify $r_B$ and increase the adverse effect of the 3D field. Here, ML-3D continuously updates the optimal coil setup in real time to minimize $r_B$ for the safe operation of the 3D field. Therefore, the successful and stable ELM suppression with strong confinement recovery is an outcome of the synergistic collaboration between $I_{RMP}$ and ML-3D feedback control, further highlighting the importance of this integrated 3D-field optimization approach. Likewise, it also has the advantage of enabling real-time 3D optimization for unexpected situations, which is essential in long-pulse plasma operations. Notably, ML-3D is based on a physics-based model and doesn't require experimental data, making its extension to ITER and future fusion reactors straightforward. This robust applicability to future devices highlights the advantage of the ML-integrated 3D-field optimization scheme. It is worth pointing out that the operational limits of the KSTAR-3D coils are restricting the ML-3D's ability to further optimize $r_B$ in #31873. In future devices with higher current limits for 3D coils, better field optimization and improved fusion performance are expected.

As shown in Fig. 5b, the plasma performance significantly decreases from $G \sim 0.17$ (4 s, before 3D-field application) to $G \sim 0.1$ after the first ELM suppression at 6.2 s, which is the major disadvantage of

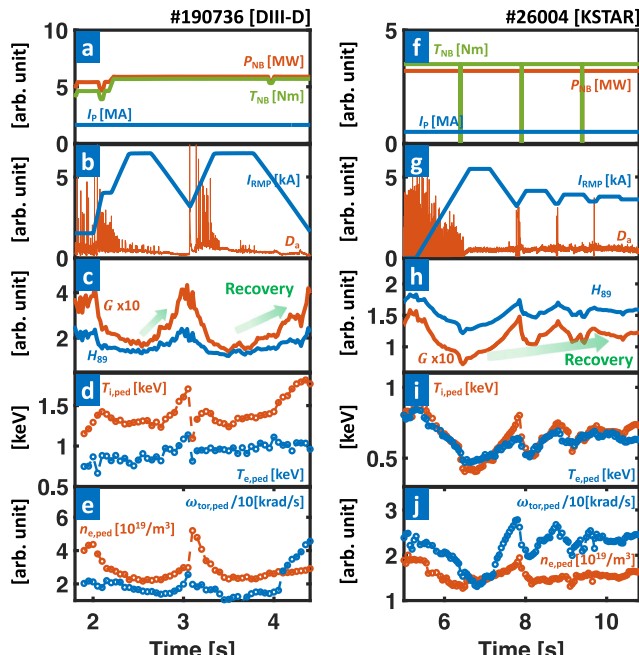

**Fig. 6 | Plasma parameters for an ELM suppression discharge (#190736 [a–e] and #26004 [f–j]) with adaptive amplitude optimization. a, f** Plasma current ($I_P$, blue), NBI heating ($P_{NB}$, orange), and torque ($T_{NB}$, green). **b, g** RMP coil current ($I_{RMP}$, blue) and particle recycling light ($D_\alpha$ emission, orange) near outer divertor target. **c, h** Normalized confinement time ($H_{89}$, blue) and figure of merit ($G$, orange). **d, i** Pedestal height of electron ($T_{e,ped}$, blue) and ion ($T_{i,ped}$, orange) temperature. **e, j** Pedestal height of electron density ($n_{e,ped}$, orange) and toroidal rotation frequency of carbon (6+) impurity ($\omega_{tor,ped}$, blue). The green arrows in **c**, **h** highlight the confinement recovery by optimization.

3D-field. Here, $G = \beta_N H_{89}/q_{95}^2$ is the figure of merit representing normalized fusion performance, $\beta_N$ is the normalized beta, $H_{89} = \tau_{exp}/\tau_{89}$ is the energy confinement quality compared to the standard tokamak plasmas, $\tau_{exp}$ is the experimental energy confinement time, and $\tau_{89}$ is the empirically derived confinement time using standard tokamak plasma database[7]. Following the initial degradation, however, the confinement starts to increase, eventually reaching a converged state by 8.7 s with an enhanced final state of $G \sim 0.16$, reaching the initial high-confinement state. This corresponds to a 60% boost from $G$ in a standard ELM-suppressed state. Such a notable fusion performance boost is an outcome of adaptive amplitude ($I_{RMP}$) optimization starting at 6.2 s, which will be described in the next section.

### Enhanced fusion performance using adaptive optimization

Figure 6 presents a compelling illustration of H-mode plasmas from both DIII-D ($n_{RMP} = 3$) and KSTAR ($n_{RMP} = 1$), effectively achieving fully suppressed ELMs through adaptive feedback RMP amplitude optimization. The RMP-hysteresis from the plasma response is harnessed in these discharges, allowing for sustained ELM suppression with lower RMP strength than initially required to access the ELM suppression regime[53]. As the RMP amplitude is reduced, the pressure pedestal height increases, leading to a notable global confinement boost in an ELM-suppressed state. In this section, we employ a pre-set RMP waveform or 3D spectrum and apply real-time feedback to control its amplitude ($I_{RMP}$). Therefore, the results illustrate the pure effect of adaptive amplitude optimization.

In DIII-D discharge (#190736) with $I_p = 1.62$ MA, $q_{95} \sim 3.35$, and ~5.8 MW of neutral beam injection heating, the plasma exhibits initial performance of $G \sim 0.39$ and $H_{89} \sim 2.15$, closely aligned with the target of the ITER baseline scenario, including plasma shape.

However, after the first stable ELM suppression is achieved through conventional RMP-ramp up ($n_{RMP} = 3$), the plasma performance notably decreases to $G \sim 0.18$ and $H_{89} \sim 1.45$. This 54% reduction in $G$ is mainly attributed to the degradation in density and temperature pedestals, as depicted in Fig. 6d, e. Similarly, in the KSTAR discharge with $I_p = 0.51$ MA, $q_{95} \sim 5$, and ~3 MW of neutral beam heating, significant performance degradation is observed from $G \sim 0.19$ and $H_{89} \sim 2.24$ to $G \sim 0.11$ and $H_{89} \sim 1.69$ after ELM suppression by $n_{RMP} = 1$ RMPs (Fig. 6h). These extensive degradations are a well-known general trend in RMP experiments[28,54–56]. Such $H_{89}$ and $G$ degradation cannot be accepted in future fusion reactors due to the substantial deviation from the ITER baseline level ($H_{89} = 2$, $G = 0.4$)[6] and the increase in fusion cost.

Following the initial degradation, the real-time adaptive RMP optimization scheme improves fusion performance while maintaining stable ELM suppression. The controller relies on the $D_\alpha$ emission signal near the outer divertor target to monitor the ELM events. To achieve ELM suppression, the RMP amplitude ($I_{RMP}$) is increased until ELM suppression. Subsequently, during the ensuing ELM-suppressed phase, the controller lowers $I_{RMP}$ to raise the pedestal height until ELMs reappear, at which point the control ramps up the RMP amplitude again to achieve suppression (Fig. 6b). A 0.5 s RMP flattop interval (longer than five times of energy confinement time) is introduced between the RMP-ramp up and down phases in the experiment to achieve a saturated RMP response. As mentioned earlier, the 3D shape of RMP is pre-programmed for safe ELM suppression and only adjusts the amplitude.

With the adaptive RMP optimization, the plasma performance of discharge #190736 is enhanced to $G \sim 0.33$ and $H_{89} \sim 2.05$, which corresponds to 83% and 41% improvement of $G$ and $H_{89}$ of standard RMP-ELM suppressed state, respectively. Notably, the increase in $G$ is particularly significant, reaching the ITER-relevant level, highlighting the advantage of adaptive optimization. We note that the further performance increase during the transient period (>2.95 s) of rapid density increase with ELM-induced sawteeth is not considered to avoid overestimating the control performance. The improved confinement quality is attributed to enhanced temperature and density pedestals. As shown in Fig. 6d, e, all pedestals are improved compared to the initial ELM suppression phase. For example, the electron ($T_{e,ped}$) and ion ($T_{i,ped}$) temperature pedestals increase by 25% and 28%, respectively. The electron density pedestal ($n_{e,ped}$) also shows a 23% increase during the same period.

A strong performance boost is similarly achieved in KSTAR discharge #26004. To leverage the long-pulse feasibility (>10 s) of KSTAR, the adaptive optimization scheme is implemented with the lower bound of $I_{RMP}$ set slightly higher (by 0.1 kA) than where the most recent ELM returns. This adaptive constraint reduces control oscillation and enables the plasma to converge to an operating point after sufficient iterations, optimizing both ELM stability and confinement. In the selected discharge, the adaptive scheme reaches a stable ELM-suppressed phase after 10 s, with enhanced global confinement, as illustrated in Fig. 6g, h. The plasma performance in this final state shows $G \sim 0.15$ and $H_{89} \sim 1.98$, increasing up to 37% and 17% of $G$ and $H_{89}$ at initial ELM suppression. This successful iteration of the adaptation scheme in longer pulses also supports its applicability in ITER long pulses.

The adaptive scheme has been extensively tested in both tokamaks over 30 discharges with multi-toroidal wave number of RMP ($n_{RMP}$) of $n_{RMP} = 1 - 2$ (KSTAR) and 3 (DIII-D), demonstrating its robust performance in boosting ELM-suppressed plasma performance. It is noteworthy that ITER-tokamak will utilize high-$n$ ($n_{RMP} = 3$) while fusion reactors may rely on low-$n$ due to engineering limitations[57]. Therefore, it is important to confirm the multi-$n$ capability of the adaptive RMP scheme. As shown in Fig. 7, we observe an effective $G$ enhancement from the standard ELM-suppressed state regardless of

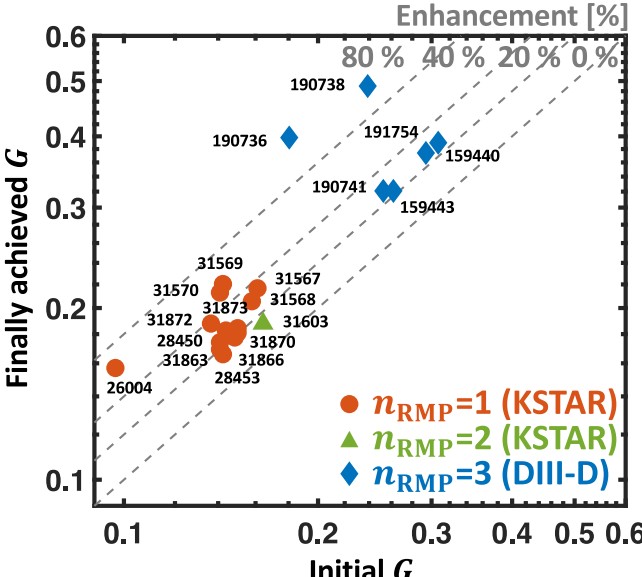

**Fig. 7 | Performance enhancement in discharge with adaptive RMP optimization.** Figure of merit ($G$) at initial (standard) ELM-suppressed state versus finally achieved $G$ from the initial state by adaptive RMP optimization. The circle (orange), triangle (green), and diamond (blue) dots correspond to $n_{RMP} = 1$ (KSTAR), $n_{RMP} = 2$ (KSTAR), and $n_{RMP} = 3$ (DIII-D) cases, respectively. Here, $n_{RMP}$ is the toroidal periodicity of RMP. The dotted gray lines show the degree of $G$ enhancement by the adaptive scheme.

$n_{RMP}$, affirming the multi-$n$ compatibility of the adaptive RMP optimization for ITER and future fusion reactors. With such success, the ELM-suppressed discharges with RMP optimization perform the best $G$ among the various ELM-free scenarios (see Fig. 2) in DIII-D and KSTAR, including Non-ELM scenarios[56] where ELMs are intrinsically suppressed without using 3D-fields. This highlights that adaptive 3D-field optimization is one of the most effective ways to achieve a high-performance ELM-free scenario. Furthermore, the enhanced $H_{89}$ can result in an increased non-inductive current fraction. This improvement reduces the flux consumption in the central solenoid, thereby extending the pulse length. Therefore, the adaptive RMP scheme has contributed to notable ELM-suppression long-pulse records[58] over 45 s in KSTAR, which is also an essential advantage for ITER operations. We emphasize that the feasibility of utilizing RMP-hysteresis in a feed-forward approach is restricted. This limitation stems from the challenges in precisely predicting the required RMP strength to achieve and sustain ELM suppression. Notably, such an advantage remains exclusive to the adaptive real-time scheme.

Interestingly, a very high $G$ boost over 80% is observed in the $n_{RMP} = 3$ results for the DIII-D cases, recovering most of the performance lost by RMP (see Fig. 7). This further highlights the performance of adaptive RMP optimization, a key to accessing ELM-suppressed high-confinement scenarios. We'll revisit the analysis and insights behind these strong performance enhancements in the last part of this paper.

### Nearly complete ELM-free operation with high performance by integrated RMP optimization

It is worth pointing out that the amplitude optimization process results in multiple ELMs before accessing the optimized state. As shown in Fig. 6b, g, the ELMs reappear during RMP amplitude optimization. These can be considered acceptable as they can be reduced with control tuning, and also, few ELMs are tolerable in future fusion machines, such as ITER[59]. However, avoiding extensive ELMs between the LH transition and the first ELM suppression is vital. Previous

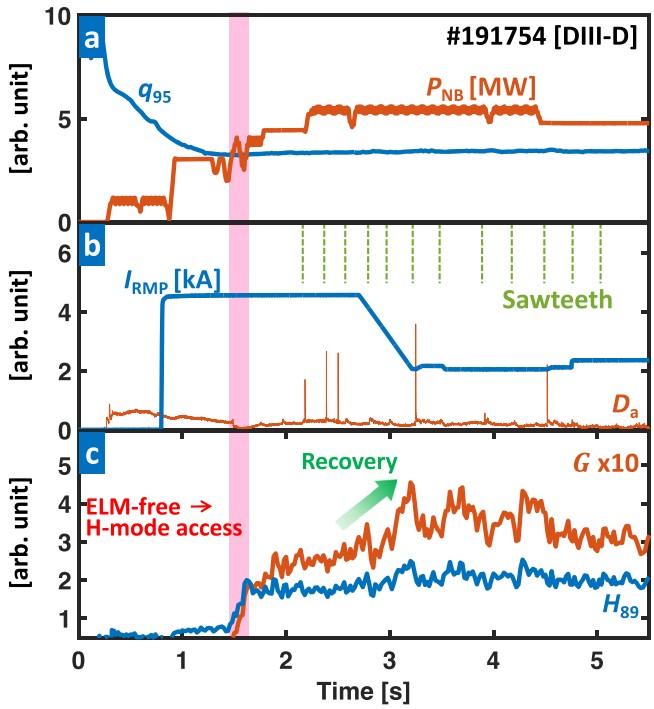

**Fig. 8 | Plasma parameters with an integrated RMP optimization (#191754), reaching near-zero ELMs. a** Edge safety factor ($q_{95}$, blue) and NBI heating ($P_{NB}$, orange). **b** RMP coil current ($I_{RMP}$, blue) and particle recycling light ($D_\alpha$ emission, orange) near outer divertor target. **c** Normalized confinement time ($H_{89}$, blue) and figure of merit ($G$, orange). The red-colored area highlights the H-mode access without early ELMs. The green dotted lines in **b** show the sawteeth timing. The green arrow in **c** highlights the confinement recovery by RMP optimization.

research has demonstrated that early RMP-ramp up[60,61] before the first ELM reduces ELMs during the early H-mode phase. Nevertheless, this approach often faced limitations due to uncertainties in determining the required conditions, including initial RMP amplitude for suppressing the first ELMs. While using a sufficiently large RMP could guarantee early ELM suppression, it leads to poor confinement.

The integration of early RMP and 3D-field optimization schemes provides an effective solution to address these limitations. Figure 8 illustrates a DIII-D discharge (#191754) of near-zero ELMs, where the adaptive RMP optimization is integrated with early RMP ramp-up. Notably, establishing a strong RMP of $I_{RMP} = 4.7$ kA (at 0.8 s) successfully suppresses early ELMs, enabling ELM-free access to H-mode. Subsequently, the RMP optimization improves the performance from 2.7 s, leading to the boost of $G = 0.28$ and $H_{89} = 1.83$ at standard ELM-suppressed state to $G = 0.39$ and $H_{89} = 2.18$. Despite the successful integration of optimizing schemes, complete ELM suppression remains challenging due to a few sporadic ELMs induced by sawteeth activity during the ELM suppressed phase, as shown in Fig. 8b. These sporadic ELMs lead the controller to overestimate the ELM instability, thereby hindering further optimizations (or decreases) of the RMP amplitude, ultimately limiting the additional improvement in confinement. Nevertheless, the plasma performance still exceeds the ITER-relevant baseline ($H_{89} = 2$)[62], highlighting the benefits of the adaptive scheme. In the future, enhancing the ELM detection algorithm with new diagnostics, such as divertor thermoelectric currents[63] or fast profile diagnostics, will be needed to separate the sawteeth effect during the optimization process for improved performance. This will also be beneficial for future devices with metallic walls, where the $D_\alpha$ signal may not be efficient for ELM detection[26]. Furthermore, additional progress can be pursued by exploring scenarios with reduced or

mitigated sawteeth, potentially leading to even greater improvements in ELM control and optimization performance.

## Physics behind on accessing highly enhanced edge-energy-burst-free phase by adaptive optimization

The achievement of the ELM-suppressed state by RMP is generally understood to be due to field penetration and pedestal gradient reduction. When RMP is applied externally, the plasma response mainly shields it, and a sufficiently strong amplitude is required to penetrate the plasma and form magnetic islands that cause additional pedestal transport. The plasma flow ($\omega_E$), formed by ExB forces due to electric ($E$) and magnetic ($B$) fields, is known to strengthen the RMP shielding effect, causing the amplitude threshold ($I_{RMP,th}$) required to access and maintain the ELM-suppressed state to increase. In particular, it is found that the value of $\omega_E$ on the rational surfaces near the electron pedestal top mainly increases $I_{RMP,th}$ because magnetic islands on these surfaces are key to the ELM suppression[64]. Following the penetration of RMPs, the pedestal gradient decreases due to RMP-induced transport, and ELM suppression is attained once the gradient falls below the ELM stability limit. In theory, the pressure gradient at the pedestal center should stay under the stability limit to avoid the reappearance of ELMs[65]. Here, this gradient reduction results in a decrease in pedestal height and global confinement. Considering these factors, strict control boundaries exist for the RMP amplitude and pedestal gradient to ensure stable ELM suppression. These limitations often constrain the strong confinement to boost through adaptive RMP optimization. Remarkably, however, the highly optimized cases exhibiting more than an 80% $G$ enhancement, as shown in Fig. 7, offer an insight to overcome limitations in a performance boost.

Figure 9 shows an ELM-suppressed discharge in DIII-D (#190738), which achieves >80% $G$ enhancement by adaptive $n = 3$ RMP optimization. After the first stable ELM suppression at 2.45 s with $I_{RMP} = 5.4$ kA, the plasma performance improves from $G \sim 0.22$ and $H_{89} \sim 1.58$ up to $G \sim 0.49$ and $H_{89} \sim 2.42$ at 3.55 s. This significant performance boost is characterized by a gradual change that differs from the transient confinement increase typically observed in transient ELM-free periods (>3.7 s) in that a sharp increase in density pedestal is not observed before 3.65 s. In these highly enhanced states, the ELM suppression is maintained until $I_{RMP} \sim 1.5$ kA, exhibiting more than 70% of RMP-hysteresis, as shown in Fig. 9a, which dramatically exceeds typical values (~40%) in other cases[66]. Because a smaller RMP amplitude means higher performance, such a strong hysteresis is the main contributor to performance enhancement.

The strong RMP-hysteresis observed in this experiment is correlated with the self-consistent evolution of the plasma flow in the RMP control. As shown in Fig. 9c, the toroidal rotation at the pedestal top ($\omega_{tor,ped}$) increases as the RMP decreases. Then, the increase in $\omega_{tor,ped}$ alters the momentum balance of the plasma, causing $\omega_{E,10/3}$ to decrease toward zero in the electron pedestal top region, located at the $q = 10/3$ rational surface. Figure 9c, e shows this correlation between increasing $\omega_{tor,ped}$ and $\omega_{E,10/3} \to 0$. As a result, $I_{RMP,th}$ is relaxed, and the RMP amplitude can be further reduced. Here, an additional decrease in RMP weakens the rotation damping by the 3D field, resulting in a further increase in $\omega_{tor,ped}$, and allows $\omega_{E,10/3}$ and $I_{RMP,th}$ to decrease favorably once again. This synergy between $I_{RMP,th}$ and $\omega_{tor,ped}$ is key to maintaining ELM suppression with very low RMP, leading to a strong confinement enhancement (and rotation), as shown in Fig. 9a–d. The ELM suppression (~4.2 s) in Fig. 6b with a very low RMP (1.5 kA) also shares the same feature. We note that achieving such a reinforced RMP-induced hysteresis is not trivial in the experiment, requiring pre-programmed and dedicated RMP waveforms. In this respect, adaptive RMP optimization is an effective methodology, as it can automatically generate and utilize the hysteresis without manual intervention.

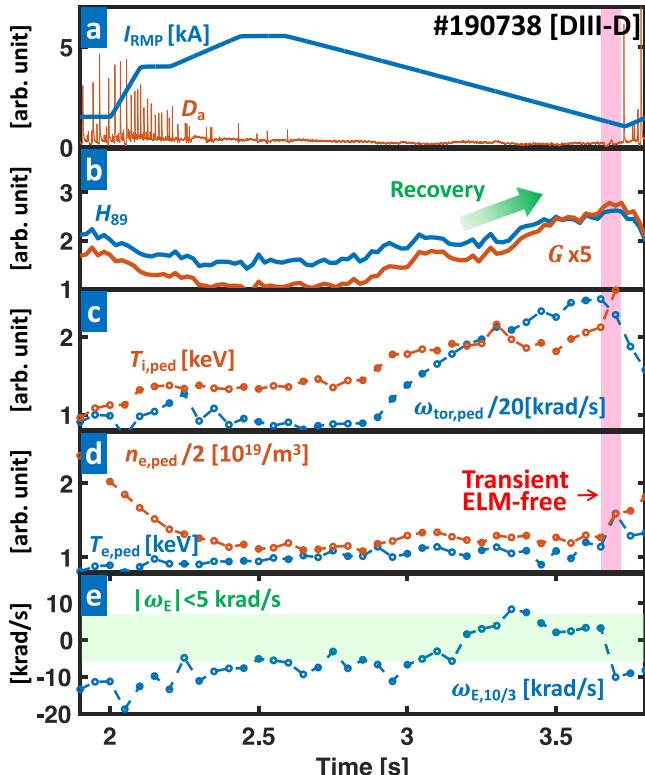

**Fig. 9 | Plasma parameters of an optimized RMP amplitude (#190738) with highly enhanced performance. a** $I_{RMP}$ (blue) and particle recycling light ($D_\alpha$ emission, orange) near the outer divertor target. **b** Normalized confinement time ($H_{89}$, blue) and figure of merit ($G$, orange). **c** Toroidal rotation frequency of carbon (6+) impurity ($\omega_{tor,ped}$, blue) and Pedestal height of ion temperature ($T_{i,ped}$, orange). **d** Pedestal height of electron temperature ($T_{e,ped}$, blue) and density ($n_{e,ped}$, orange). **e** $E \times B$ rotation frequency ($\omega_E$, blue) at $q = 10/3$, where $q$ is a safety factor. The red and green colored regions show the transient ELM-free phase and $|\omega_E| < 5$ krad/s, respectively. The green arrow in **b** highlights the confinement recovery by RMP optimization. A radial profile evolution for this discharge can be found in Supplementary Fig. 2.

The enhanced RMP-hysteresis and rotation increase observed in the experiments also offer promising aspects for future fusion devices. Maintaining thermal and energetic particle confinements in a fusion reactor is essential for achieving high fusion performance ($G$). However, the presence of RMPs leads to undesired perturbed fields in the core region that adversely affect the fast ion confinement. Additionally, RMP-induced rotation damping poses a critical challenge for ITER, where externally driven torque may not be sufficient to suppress core instabilities and turbulent transport. The strengthened RMP-hysteresis and rotation boost during adaptive RMP optimization can significantly mitigate these unfavorable aspects of RMPs by enabling ELM suppression with very low RMP amplitudes. By reducing the negative impacts of RMPs on fast ion and core confinements, the prospects of an adaptive scheme for achieving high fusion products within future fusion devices become more favorable.

It is noteworthy that the $\omega_{tor,ped}$ increase in the early RMP-ramp down phase still leaves a question. This may simply be due to the reduced damping caused by the 3D fields[67,68]. However, the increase in $\omega_{tor,ped}$ starts 0.3 s later than the 2.6 s that the RMP-ramp down starts. In that the ELM dynamic and rotation response are weakly correlated here, this delayed response may indicate that additional mechanisms, such as field penetration or turbulence, are participating in the rotation response. In fact, the change in turbulence along with rotation change is also observed in the experiment. Future studies on plasma rotation in the presence of RMPs will provide further insight

into the projection of the RMP-ELM scenario onto ITER and future devices.

## Discussion

We have successfully optimized controlled ELM-free states with highly enhanced fusion performance in the KSTAR and DIII-D devices, covering low-$n$ RMPs relevant for future reactors to ITER-relevant $n_{RMP} = 3$ RMPs and achieving the highest $G$ among various ELM-free scenarios in both machines. Furthermore, the innovative integration of the ML algorithm with RMP control enables fully automated 3D-field optimization and ELM-free operation for the first time with strong performance enhancement, supported by an adaptive optimization process. This adaptive approach exhibits compatibility between RMP ELM suppression and high confinement. Additionally, it provides a robust strategy for achieving stable ELM suppression in long-pulse scenarios[58] (lasting more than 45 s) by minimizing the loss of confinement and non-inductive current fraction[69]. Notably, a remarkable performance ($G$) boost is observed in DIII-D with $n_{RMP} = 3$ RMPs, showing over a 90% increase from the initial standard ELM-suppressed state. This enhancement isn't solely attributed to adaptive RMP control but also to the self-consistent evolution of plasma rotation. This response enables ELM suppression with very low RMP amplitudes, leading to enhanced pedestal. This feature is a good example of a system that transitions to an optimal state through a self-organized response to adaptive modulation. In addition, the adaptive scheme is integrated with early RMP-ramp methods, achieving an ITER-relevant ELM-free scenario with nearly complete ELM-free operation. These results confirm that the integrated adaptive RMP control is a highly promising approach for optimizing the ELM-suppressed state, potentially addressing one of the most formidable challenges in achieving practical and economically viable fusion energy.

Such an effective application of this control method across two tokamaks and ELM-suppression scenarios shows its robust compatibility with plasma operation that satisfies conditions for ELM suppression. However, its efficiency in future fusion reactors deserves further investigation, closely linked to the accessibility of the ELM suppression state. Previous studies have shown that this accessibility requires specific plasma conditions[63,70], which can impose constraints on reactor design and introduce uncertainty about whether these conditions can be achieved. Recently, a detailed modeling study[65] suggests that these conditions are attainable in ITER, identifying the RMP as one of the leading approaches for ELM control in its design[11,71]. Therefore, when scenario accessibility is achieved through these design efforts, the control strategy will remain effective. Ongoing progress in theoretical and modeling research will continue to enrich our understanding of its application in future fusion energy systems.

In future fusion reactors like ITER, the presence of metallic walls may introduce new challenges for implementing this control strategy due to the limited experience with operational regimes in current nonmetallic devices. Using high-$Z$ metallic walls could lead to performance issues and core instabilities caused by impurity accumulation[72]. While RMP drives additional impurity transport at the pedestal, mitigating impurity accumulation[26] to some extent, careful consideration of impurity accumulation with RMP remains essential for ITER. Here, adaptive control can be extended to provide the optimal RMP that reduces the accumulation through balancing the transport[73], source[74,75], and penetration of these impurities[76]. This will ensure effective impurity removal and preserve high plasma confinement with ELM suppression, guided by continuous monitoring of impurity levels and ELM dynamics. In future work, these uncertainties and potential solutions will be explored through advanced modeling and further demonstration of the control method for existing and upcoming tokamaks with metallic walls.

Lastly, there are remaining features that need to be enhanced to achieve fully adaptive RMP optimization over the entire discharge in

future devices. Current strategies reliant on ELM detection encounter several ELMs during optimization, which is not ideal for fusion reactors where minimizing potential risk is crucial. Earlier research[66] has revealed a precursor pattern in $D_\alpha$ and turbulence. This distinctive pattern can be harnessed for real-time preemptive RMP adjustments to prevent ELM occurrences, ultimately achieving complete ELM-free optimization. The initial test on KSTAR has shown promising results for this concept, almost entirely suppressing ELMs by tracking precursors in $D_\alpha$ signals[77]. For future fusion devices, improved ELM control will be enabled with the advancement of methods to detect ELM-loss precursors, incorporating measurements of high-frequency fluctuations in real-time. The integration of ML algorithms in real-time signal processing will be crucial for such effective pattern recognition. Additionally, achieving stable H-mode operation without the occurrence of the first ELM, as demonstrated in Fig. 8b, is an additional challenge in fully adaptive optimization. It has been observed that careful adjustment of $q_{95}$ and early RMP application before entering H-mode is key to suppressing the first ELM. Here, early initiation of RMP could potentially hinder the transition to H-mode in future reactors[78-83]. Addressing this issue will necessitate fine-tuned early plasma scenarios and RMP ramp timings to mitigate their impact[61].

In conclusion, RMP with new real-time control and ML techniques shows a promising path for optimizing ELM control to support its application in ITER and ongoing future device design. Continued research and development on remaining questions, as well as the improvement of alternative ELM-free scenarios, will develop broad, robust, and advanced ELM control solutions for ITER and future tokamaks.

## Methods

### DIII-D tokamak
The DIII-D tokamak is the largest operating national tokamak device in USA. The reference discharge has the plasma major radius $R_0 = 1.68$ m, minor radius $a_0 = 0.59$ m, and the toroidal magnetic field $B_T = 1.92$ T at major radius $R_0$. The $n = 3$ RMP ELM suppression discharge is reproduced with a plasma shape having elongation $\kappa \sim 1.81$, upper triangularity $\delta_{up} \sim 0.35$, and lower triangularity $\delta_{low} \sim 0.69$.

### KSTAR tokamak
The KSTAR tokamak is the largest magnetic fusion device in the Republic of Korea, supported by the Korea Institute of Fusion Energy (KFE) and Government funds. The reference discharge has the plasma major radius $R_0 = 1.8$ m, minor radius $a_0 = 0.45$ m, and the toroidal magnetic field $B_T = 1.8$ T at major radius $R_0$. The $n = 1$ RMP ELM suppression discharge on KSTAR can be reproduced with a plasma shape having elongation $\kappa \sim 1.71$, upper triangularity $\delta_{up} \sim 0.37$, and lower triangularity $\delta_{low} \sim 0.85$.

### ELM-free database
The ELM-free database in DIII-D tokamak comes from ref. 56. Here, the previous database uses 300 ms time averaging, while the data point of discharge with adaptive RMP optimization uses a shorter time scale (100 ms) to capture the performance variation with adaptive RMP optimization. The KSTAR database is also constructed using the same process. In this work, the RMP optimization database covers $q_{95} = 3.3 - 5$, $n_{e,ped} = 1.5 - 3 \times 10^{19} / m^3$, and heating power of 2–6 MW.

### Ideal plasma response calculation
The perturbed radial fields ($\delta B_r$) from an ideal plasma response by RMP are calculated using GPEC code[49] under given magnetic equilibria and 3D coil configuration. The core ($B_{core}$) and edge ($B_{edge}$) responses are derived through radially averaging $\delta B_r$ at $\psi_N = 0 - 0.9$ and $0.9 - 1.0$, respectively. Here, the $q = 1$ surface is removed during the calculation to exclude the 1/1 resonant effect. The optimal 3D coil configurations

($R_M$, $R_B$, $\phi_T$, $\phi_B$) of the edge-localized-RMP model are derived using calculated perturbed fields.

### Surrogate 3D model
The surrogate model is developed using the dense layer model within the Keras library. The hidden neurons are equipped with ReLU activation function, and they are organized in two layers with 40 and 10 neurons, respectively. In order to train this model, we collected data from 8490 KSTAR time slices in the past three years. The data was split randomly into 6790 and 1700 samples for training and testing the model. In total, this MLP consists of 800 trainable parameters (connection weights), and the training iterations continue for 150 epochs or if the error rates converge. The final R2 score on the test set is 0.91.

### Kinetic profile and equilibria reconstruction
Core ion temperature is measured by charge exchange recombination system[84,85] for carbon (6+) impurities at outboard mid-plane. Core electron temperature and density are measured by the Thomson scattering system[86-88]. To obtain well-resolved profiles, the data are averaged over 50 ms. The pedestal height is obtained from hyperbolic tangent fits with edge profiles. Kinetic equilibria are reconstructed for the plasma transport and stability analysis. This equilibrium is calculated from the magnetic reconstruction using EFIT code[89] with the reconstructed radial profiles. The OMFIT package[90,91] is used to achieve well-converged equilibrium with automated iteration processes.

### Plasma fluctuation measurements
In this work, edge $n_e$ fluctuations are measured from the Doppler backscattering system[92]. Here, the measured density fluctuation captures the ion-scale turbulence $k_y\rho_s = 0.3 - 1.5$, rotating in the electron direction, where $k_y$ is the bi-normal wave number, $\rho_s = \sqrt{2m_iT_e}/eB$ is the hybrid Larmor radius, and $m_i$ is deuterium mass.

## Data availability
Raw data were generated from the DIII-D and KSTAR teams. The data supporting the findings of this work are available from the corresponding author upon request.

## Code availability
Source codes were developed by authors. The access can be available from the corresponding author upon request.

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

## Acknowledgements

The authors would like to express their deepest gratitude to KSTAR and the DIII-D team. This material was supported by the U.S. Department of Energy, under awards DE-SC0020372, DE-SC0024527, DE-AC52-07NA27344, DE-AC05-00OR22725, DE-FG02-99ER54531, DE-SC0022270, DE-SC0022272, and DE-SC0019352. The U.S. Department of Energy also supported this work under contract number DEAC02-09CH11466 (Princeton Plasma Physics Laboratory). The United States Government retains a non-exclusive, paidup, irrevocable, and worldwide license to publish or reproduce the published form of this manuscript or allow others to do so for United States Government purposes. This material is based upon work supported by the U.S. Department of Energy, Office of Science, Office of Fusion Energy Sciences, using the DIII-D National Fusion Facility, a DOE Office of Science user facility, under award(s) DE-FC02-04ER54698. This research was also supported by the R&D Program of "KSTAR experimental collaboration and fusion plasma research (EN2401-15)" through the KFE, funded by government funds, and the Technology development projects for Leading Nuclear Fusion through the National Research Foundation of Korea (NRF) funded by the Ministry of Science and ICT (No. RS-2024-00281276). This report was prepared as an account of work sponsored by an agency of the United States Government. Neither the United States Government nor any agency thereof, nor any of their employees, makes any warranty, express or implied, or assumes any legal liability or responsibility for the accuracy, completeness, or usefulness of any information, apparatus, product, or process disclosed, or represents that its use would not infringe privately owned rights. Reference herein to any specific commercial product, process, or service by trade name, trademark, manufacturer, or otherwise does not necessarily constitute or imply its endorsement, recommendation, or favoring by the United States Government or any agency thereof. The views and opinions of authors expressed herein do not necessarily state or reflect those of the United States Government or any agency thereof.

## Author contributions

S.K.K., R.S., and S.M.Y. led the control development, experimental demonstration, and analysis. E.K. conceived the original idea of adaptive control. Q.H. gave and shared the fundamental guidance of the

experimental plan and analysis. A.O.N. analyzed the micro instability with Gyro-kinetic code. S.H.H., R.W., N.C.L., Y.M.J., M.W.K., and A. Battey supported the experimental procedures. R.N. and C.P.S. discussed the critical physics picture of transports at the pedestal. J.K.P. and N.C.L. discussed the role of RMP response, stability, and transport analysis of the pedestal region. R.H., T.R., G.Y., W.H.K., and J.H.L. analyzed the profile and fluctuation measurements. A.J. supported the development of the surrogate model. Y.-S.N., A. Bortolon, and J.S. advise the scope of the experimental analysis. S.K.K. wrote the main manuscript text, and A.O.N. and all authors reviewed it.

## Competing interests

The authors declare no competing interests.
