## [Peer Review File · Nature Communications]

Highest Fusion Performance without Harmful Edge Energy Bursts in TokamakReviewer #1 (Remarks to the Author):

Referee report

Article ID: NCOMMS-23-50991

Title: Highest Fusion Performance without Harmful Edge Energy Bursts in Tokamak.

The paper focuses on improving plasma stability in tokamak fusion reactors. It addresses the challenge of transient energy bursts, known as Edge Localized Modes (ELMs), which can be detrimental to reactor components. The study presents an innovative approach using 3D magnetic perturbations, optimized through machine learning techniques, to suppress these instabilities. This method, successfully tested on DIII-D and KSTAR tokamaks, achieves high fusion gain without triggering potentially damaging instabilities. The paper emphasizes the importance of this approach for future reactors such as ITER, highlighting its role in enhancing plasma confinement and maintaining stability.

The paper is well written and addresses one of the most important concerns for the exploitation of fusion reactors as a core component of future power plants. As far as the methodology is concerned, robustness and reproducibility of the proposed techniques are relatively well assessed and experimentally tested.

Nevertheless, I have several remarks regarding the following points:

1) The insufficient description of the many studies on different ELM regimes under reactor relevant conditions, as well as the plethora of techniques available to mitigate the high heat fluxes associated to ELMs.

Despite the great achievements of recent no-ELMs or small-ELMs scenarios, (such as EDA H-mode QCE H-mode, QHM, etc.), well demonstrated in various devices (C-Mod, and more recently in EAST, AUG, and DIII-D itself) critical challenges remain to be solved.

One of the prime concerns is the fact that the pedestal collisionality in present-day machines is higher than what is required for a fusion reactor. And there are several physics questions like high density, low impurity content, compatibility with tungsten walls, possibility of access at low input torque and power, with dominant electron heating, no impurity accumulation despite the absence of ELMs (see following point related to disruptions). Although the paper is not a review of the various (no-)ELMs regimes, it is worth mentioning that this is not the end of the story and that, but there are still several open questions regarding the extrapolation of these ELM suppression techniques to large scale devices under reactor relevant conditions. Finally, the experimental program of many fusion facilities will continue to investigate alternative Type-I ELM regimes, some of which still lack theoretical basis and, moreover, routine operation in those regimes has yet to be demonstrated.

2) play an important role in metallic devices to regulate the transport of high-Z impurities.

The main reference to disruptions in the paper is related to the need of minimizing the potential core perturbation associated with the edge 3D fields required for ELMs suppression, as this can potentially cause core instability which may eventually lead to disruptions. One of the main problems in high-performance scenarios in metal wall devices, is the avoidance of high-Z impurities accumulation, which can be prevented with frequent small ELMs, which help to regulate their transport and to flush them out. In high power and high current discharges (see JET special contribution at the last IAEA-FEC 2023 for instance), even a short ELM-free phase can immediately trigger the accumulation of impurities, leading to an immediate degradation of the performance and eventually to disruptions. This aspect is very important and should be properly discussed .

3) The pipeline implemented for the automated 3D-fields optimization, especially concerning the role of machine learning in the optimization.

Successful suppression experiment of the ELM crash with a priori modeling with the optimal phase and amplitude of the RMP coil current was already successfully carried out in KSTAR [J.K. Park Nature Physics (2018)].

As stated in this paper, "This approach exploits the physics-based optimization scheme of 3D

waveform based on plasma equilibrium and ideal 3D response from GPEC simulation.... its computation time, which takes tens of seconds, hinders real-time applicability, limiting its use to pre-programmed or feed-forward strategies.". As far as I understand from the description in the paper, the machine learning model implemented to overcome such limitations is a surrogate model of the GPEC code (a physics-based model for 3D field optimization). The equilibrium parameters in input to the surrogate model are very likely not changing significantly in what is supposed to be the no-ELM H-mode phase, therefore they could possibly be calculated beforehand with an inverse equilibrium solver.

It appears that the RMP optimizer adaptively changes the current I_{RMP} in real-time by monitoring the ELM state using the D_α signal with a strategy that, as far as the variation of I_{RMP} is concerned, does not seem to require any feedback from the ML component of the algorithm. The feedback is based on the detection of the ELM, which does not depend on the surrogate model, because, if I understand correctly the control scheme and the examples reported, ELM suppression is achieved by a linear increase of RMP linear increase followed by a short plateau once the suppression is achieved to avoid transient oscillations. The degradation should mainly depend on the evolution of the density and temperature pedestals rather than the equilibrium parameters, so my main question is what the surrogate model can add to a pre-computed offline optimization done with the GPEC code coupled to an inverse equilibrium solver for example.

It seems that the main effect is due to the regulation of I_{RMP} while the other parameters, which do not change significantly, need to be optimized for different plasma scenarios. This also seems to be true for the minimization of the 3D fields needed for full ELMs suppression. Is this correct?

Another important question related to this point regards the variability of the scenarios in which the ELMs suppression was tested in the real-time experiments (range of heating power, etc.). I think it would be worth adding more details with respect to this point, since this would better qualify aspects such as the robustness and the extrapolability of the developed solution.

Nevertheless, I think that the work described in this paper presents several elements of novelty, it addresses a topic of great interest to the fusion community and not only. It attempts to bridge the gap between theory, simulations and experiments, taking advantage, among other things, of recent developments in machine learning. The present paper can be considered for publication in this Journal, only if the authors adequately address the issues and concerns reported here, including the "minor" points below:

- Why the choice of using H89 instead of the most recent H98y,2 plasma confinement scaling?

- How have the RMP ramp-up rates and the length of the plateau after ELMs suppression been optimized in the two machines? Is there any physics-based reason for their choice?

- Could you elaborate a bit more on the "precursor pattern" in D_α and turbulence signals, emerging about 20 ms ahead of ELM reappearances during the suppression phase? Would those turbulence markers be available in real-time? If that is the case, would their processing be feasible in real-time?

Reviewer #2 (Remarks to the Author):

The manuscript (MS) titled "Highest Fusion Performance without Harmful Edge Energy Bursts in Tokamak" describes the 3D field optimization, exploiting machine learning, real-time adaptability. The ELM mitigation and the improvement of the figures of merit are vital important for the fusion reactor in the future, so the MS is of great interesting for the plasma society; however, the MS could not be accepted as it is for publish due to the following reasons:

1) It seems the authors confused the concepts of fusion gain and figures of merit. Following the definition " $G = \beta_N \cdot H_{89} / q_{95}^2$ " in the caption of Page 4, I guess that G refers to the figures of merit. The authors should be very carefully to re-organize and clarify the fusion gain/figures of merit

in the context.

2) In Fig. 2, there is a small region of RMP (DIII-D) that does not overlap with that of Non-ELMs (DIII-D). Could the authors explain how they obtained/defined the colored region for RMP DIII-D, KSTAR and Non-ELMs for DIII-D, respectively? And please explain the (possible) reasons.

Besides, some points of the Adaptive RMP (DIII-D) locate outside the regions of RMP and Non-ELMs for DIII-D. Could the authors provide some explanations? What measures have been taken to achieve the value of G higher than 0.4?

3) In Fig. 6 (b) and Fig. 9 (a), the amplitudes of D_α are too low, it is very difficult for the readers to determine the existence of the ELM (during the flattop and decreasing phases of I_{RMP} in Fig. 9(a), some mossy ELM seems to exist) or not. Besides, in the paragraph right before the start of II. Discussion, the authors declared that the "increase in $\omega_{tor,ped}$ starts 0.3 s later than the 2.6s that RMP-ramp down starts", is it possible for the authors to exclude the influence of the ELM on the velocity of rotation?

4) In Fig. 7, the maximum enhancement of KSTAR is around 50% whereas $\sim 90\%$ for DIII-D. Could authors clarify the reason?

5) In Fig. 6 (d)~(e), the ion/electron temperature drops when the ELM occurs; meanwhile, the electron density rises abruptly; one might draw that the pressure might be constant during the ELM burst. However, this phenomenon could not be observed on KSTAR (Fig. (i) and (j)). Could the authors clarify the phenomenon?

6) Typos.

Caption of Fig. 5, c H89(blue)..... should be b H89(blue)

Caption of Fig. 8. q_{95} should be q_{95}

Caption of Fig. 9. c $T_{e,ped}$ should be d $T_{e,ped}$

Reviewer #3 (Remarks to the Author):

Introduction, key results and significance

One of the main operational hurdles for a tokamak concept of magnetic confinement fusion reactor is the avoidance of high transient heat fluxes towards the plasma-facing components that are the consequence of edge localized mode (ELMs) instabilities. Therefore, the fusion research community worldwide has dedicated a considerable effort to understand the physics behind these phenomena and to develop methods to mitigate the associated heat fluxes to acceptable levels for long-term operation of a future fusion reactor. ELM control by resonant magnetic perturbation (RMP) field is one of the first methods to have provided satisfactory results in that regard and was thus chosen as the main ELM control scheme for tokamak ITER. However, the ELM suppression by RMP comes at a cost of reduced pressure of the confined plasma, as a consequence to the increase in plasma transport caused by the stochasticization of the magnetic field lines in the edge plasma. Furthermore, the full ELM suppression is typically accessible only in a narrow plasma and RMP parameter space. Thus, recently other methods for ELM heat flux control have been considered and investigated. These include but are not limited to divertor detachment or utilization of plasma confinement regimes where ELMs are absent altogether.

In this manuscript, the authors present a solution how to significantly limit the above mentioned RMP ELM control drawbacks. They do so by utilizing a linear description of edge vs core coupling of RMP to plasma, based on ideal magnetohydrodynamic approximation that has been recently reported to achieve very successful predictive results for ELM suppression on KSTAR tokamak. Within this manuscript the authors also demonstrate this method to work on the tokamak DIII-D. The novelty of the manuscript also resides in the authors having utilized the machine learning approach to design a control algorithm to regulate the RMP field spatial distribution and magnitude - to maintain the ELM suppression during the changing conditions of tokamak discharge and to limit the necessary perturbation of plasma. Up till this point this has only been done by setting a predefined RMP current waveforms and spatial distribution before the discharge took place, hence this automation presents an

important step ahead in the ELM control. Furthermore, it is shown in the manuscript that limiting the RMP-induced plasma confinement deterioration by this method results in the level of plasma confinement that is highly competitive, if not superior, to the alternative methods for ELM control. By doing so the authors challenge the paradigm among many members of the fusion research community that ELM suppression by RMP is difficult to be reliably guaranteed in experimental practice and that the associated deterioration of the H-mode plasma confinement would be too much of a tradeoff to be viable for a reactor.

I thus recommend to the manuscript to be published pending minor revisions. Among those, my main comment relates towards the limitation of the chosen methodology for ELM detection when extrapolated to a fusion reactor (see Data and methodology).

Validity and robustness of data

The data presented by the authors appears trustworthy. Tokamaks are complex, large physical experiments operated by dozens of physicists and engineers. Although the full details of such experimental arrangement and plasma therein are challenging to encompass within a single publication, the presence or absence of ELMs is very evident within the experiment operation practice. Fig. 9 in particular is very convincing of the validity of the method and correct interpretation of the data as it correctly shows well-known degradation of plasma density due to RMP pulse, improvement in plasma temperature (a new consequence of the optimization method presented by the authors) and, most importantly, correct distinction of the ELM-suppressed regime from transient ELM-free phase where plasma density ramps up (that otherwise could have been argued to be the cause of the confinement improvement).

Data and methodology

To detect the presence or reappearance of the ELMs for the real-time optimization algorithm to adjust the RMP accordingly, the authors rely on the bursts of Da radiation signal. This signal originates from the interaction of hot plasma particles released from confined region during the ELM event and the hydrogen (isotopes) deposited on the first wall of the tokamak (typically the divertor). Such method is indeed viable for the majority of the present-day tokamaks (including KSTAR and DIII-D – the two devices utilized in the manuscript), since these typically use graphite as a material for the first wall (see e.g. [1]) that is known for high long-term retention of hydrogen. However, the authors aim to provide a robust control scheme of RMP ELM suppression to be used on future reactors ITER, DEMO and possibly beyond. ITER is presently investigating the consequences of switching from a beryllium first wall to tungsten – see the press release by the Head of the Science Division of ITER from October 2023 [2] and numerous presentations and posters by the ITER Organization at recent plasma physics conferences. One of the reasons behind this decision is tungsten having lower retention of hydrogen (tritium, in particular) – see review [3] and references therein. For this reason, the Da signal on ITER will not be as reliable for ELM detection as on present day KSTAR and DIII-D tokamaks. An example of this can be seen at tokamak ASDEX Upgrade - a device operating with tungsten-coated first wall since 2007 [4], particularly in their first report of ELM suppression by RMP [5]. Thus, in the more recent ELM suppression experiments, the ASDEX Upgrade team has relied on the divertor thermoelectric currents to detect the presence and absence of ELMs [6]. The authors should thus comment in the manuscript on the viability of the Da signal as an ELM indicator in tungsten first wall tokamaks, and on the possibility of expanding the controller of their algorithm for other, more reactor-relevant ELM detection methods.

Furthermore, the authors state in the Methods section about the ideal plasma response calculation by GPEC that the core response part extends to $\psi_N = 0.9$ all the way from $\psi_N = 0.0$. This is not generally true, since the calculation of the core coupling part by GPEC starts outside the 1/1 resonant if it is present - not at the magnetic axis of $\psi_N = 0.0$.

Clarity and context

To a general reader, the authors provide a very good introduction to the topic of plasma confinement, ELMs and their control by RMP. However, the introduction is missing a brief discussion of other alternatives of resolving the ELM heat fluxes, and their performance in comparison to the method proposed by authors - although this is discussed later, on page 6. The impact of the manuscript on its respective field would be better conveyed to a more specialized reader if either fig. 2 in the manuscript was replaced by fig. 1 from the supplement, or if the above-mentioned conclusions from page 6 were also stated in the introduction.

Other than that, I also have minor comments regarding clarity of some statements in the manuscript:

- First paragraph, page 2: "produce" is confusing and should be replaced by e.g. "achieve", as the related term "fusion triple product" does not refer to production, but to product as a result of multiplication.
- Second paragraph, page 2: It is not clear which particular "significant advantage" the authors refer to when mentioning the stabilization of the ELMs by RMP on ITER. Needs more details.
- Section I. Results, first paragraph: The relation between high B_{core} and the plasma disruption is not known to a general reader not familiar with RMP experiments on tokamaks. The authors need to elaborate more.
- Same paragraph: Single disruption terminating the life of a fusion reactor is too strong of a statement. Although it is true that the number of disruptions will have to be significantly limited in a reactor, it is likely (taking ITER as the closest example) that a disruption budget will be formulated for a reactor detailing how many disruptions of which plasma scenarios can the device withstand - see e.g. [7]. Furthermore, disruptions due to high B_{core} are typically preceded by a detectable precursor signature [8] that allows disruption mitigation mechanisms to be engaged, to ensure a smoother termination of the discharge and to limit the extent of possible damage to the machine [9].
- Page 4: Phrase "totally new... path" should be replaced by a more neutral form e.g. "alternative... path". Likewise, in the next sentence it would be better to refer to the plot, to replace "It is clear" with e.g. "from the plot it is apparent".
- Page 6: the general sentence "few ELMs are tolerable in future fusion machines" should be supported by specific a reference to e.g. the ITER budget for allowed ELM size and count.
- Discussion section, first paragraph: in "achieving the highest fusion among various ELM-free scenarios..." I assume the authors intended to write "fusion product"?

References

- [1] M. Kaufmann, R. Neu, Tungsten as first material in fusion devices, *Fusion Engineering and Design* 82, 521-527 (2007).
- [2] A. Loarte, STAC committee reviews new plans for construction and operation, <https://www.iter.org/newsline/-/3935>
- [3] R. A. Causey, Hydrogen isotope retention and recycling in fusion reactor plasma-facing components, *Journal of Nuclear Materials* 300, 91-117 (2002).
- [4] A. Kallenbach, Overview of ASDEX Upgrade results, *Nuclear Fusion* 57, 102015 (2017).
- [5] W. Suttrop, et al., First Observation of Edge Localized Modes Mitigation with Resonant and Nonresonant Magnetic Perturbations in ASDEX Upgrade, *Physical Review Letters* 106, 225004 (2011).
- [6] W. Suttrop, et al., Experimental conditions to suppress edge localized modes by magnetic perturbations in the ASDEX Upgrade tokamak, *Nuclear Fusion* 58, 096031 (2018).
- [7] M. Lehnen, et al., Plasma disruption management in ITER, Proc. 26th IAEA Fusion Energy Conference, Kyoto 2016, paper EX/P6-39 (2017).
- [8] P.C. de Vries, et al., Scaling of the MHD perturbation amplitude required to trigger a disruption and predictions for ITER, *Nuclear Fusion* 56, 026007 (2016).
- [9] M. Lehnen, et al., Disruption mitigation by massive gas injection in JET, *Nuclear Fusion* 51, 123010 (2011).

Responses to Reviewers' Comments for Manuscript NCOMMS-23-50991

Highest Fusion Performance without Harmful Edge Energy Bursts in Tokamak

Addressed Comments for Publication to

by

SangKyeun Kim, et al.

Authors' Response to Reviewer 1

General Comments.

The paper focuses on improving plasma stability in tokamak fusion reactors. It addresses the challenge of transient energy bursts, known as Edge Localized Modes (ELMs), which can be detrimental to reactor components. The study presents an innovative approach using 3D magnetic perturbations, optimized through machine learning techniques, to suppress these instabilities. This method, successfully tested on DIII-D and KSTAR tokamaks, achieves high fusion gain without triggering potentially damaging instabilities. The paper emphasizes the importance of this approach for future reactors such as ITER, highlighting its role in enhancing plasma confinement and maintaining stability. The paper is well-written and addresses one of the most important concerns for the exploitation of fusion reactors as a core component of future power plants. As far as the methodology is concerned, the robustness and reproducibility of the proposed techniques are relatively well-assessed and experimentally tested. Nevertheless, I have several remarks regarding the following points:

Response:

Thank you for the orderly summary and detailed feedback you gave to the text of this manuscript. Your thoughtful comments are greatly appreciated and have helped to make this article more accessible to a wider audience.

Comment 1

There is an insufficient description of the many studies on different ELM regimes under reactor-relevant conditions, as well as the plethora of techniques available to mitigate the high heat fluxes associated with ELMs. Despite the great achievements of recent no-ELMs or small-ELMs scenarios (such as EDA H-mode, QCE H-mode, QHM, etc.), well demonstrated in various devices (C-Mod, and more recently in EAST, AUG, and DIII-D itself) critical challenges remain to be solved. One of the prime concerns is the fact that the pedestal collisionality in present-day machines is higher than what is required for a fusion reactor. There are several physics questions like high density, low impurity content, compatibility with tungsten walls, the possibility of access at low input torque and power, with dominant electron heating, and no impurity accumulation despite the absence of ELMs (see the following point related to disruptions). Although the paper is not a review of the various (no-) ELM regimes, it is worth mentioning that this is not the end of the story. There are still several open questions regarding the extrapolation of these ELM suppression techniques to large-scale devices under reactor-relevant conditions. Finally, the experimental program of many fusion facilities will continue to investigate alternative Type-I ELM regimes, some of which still lack theoretical basis and, moreover, routine operation in those regimes has yet to be demonstrated.

Response:

Thank you for your sincere comments. As mentioned, in addition to RMP, there have been several successful efforts to develop spontaneous small or non-ELMing regimes. However, all of these methods, including RMP, are still characterized by uncertainty in their feasibility in future devices. This is because the operation regime of a fusion pilot plant (high density, low impurity content, tungsten walls, possibility of access at low input torque) cannot be fully covered by existing devices. Addressing this gap remains a challenge within the fusion community and has to be investigated.

This is also connected to the referee's 2nd comment about the high-Z (tungsten) impurity issue, which severely deteriorates plasma performance. While RMP drives additional impurity transport at the pedestal, mitigating this detrimental impurity accumulation, careful consideration of impurity accumulation with RMP is vital for ITER.

Here, this new 3D control scheme can be extended to handle the high-impurity accumulation. Applying RMP can increase neoclassical impurity radial transport and decrease tungsten sources caused by ELMs, thereby reducing core accumulation. However, it is essential to navigate a delicate balance, as excessive RMP decreases both the ELM size and the ion pedestal, leading to easier high-Z impurity penetration into the core plasma and undesired confinement degradation. Therefore, real-time control schemes are imperative to sustain a minimal but sufficient I_{RMP} for radial impurity pumping while avoiding adverse effects. Simultaneously monitoring the impurity accumulation and ELM property will enable the determination of the optimal I_{RMP} for achieving high confinement with controlled impurity concentration. The future integration of these features, as well as the improvement of alternative ELM-free scenarios, can promise broad, robust, and advanced ELM control solutions for ITER and future tokamaks.

To address this referee's comment,

- 1) Small and non-ELMing regimes are discussed in the introduction (page 2, 2nd paragraph), with references for EDA/I/QH modes and small ELM regimes, including QCE. The authors think this will also help readers understand the performance comparison provided in Figure 2 in the manuscript. We have added references here as well.
- 2) Text describing the above discussions on the remaining challenges in future devices and impurity control has been added in the last paragraph of the Discussion section (page 9, 2nd paragraph).

Comment 2

Play an important role in metallic devices to regulate the transport of high-Z impurities. The main reference to disruptions in the paper is related to the need to minimize the potential core perturbation associated with the edge 3D fields required for ELM suppression, as this can potentially cause core instability, which may eventually lead to disruptions. One of the main problems in high-performance scenarios in metal wall devices is the avoidance of high-Z impurities accumulation, which can be prevented with frequent small ELMs, which help regulate their transport and flush them out. In high power and high current discharges (see JET's special contribution at the last IAEA-FEC 2023, for instance), even a short ELM-free phase can immediately trigger the accumulation of impurities, leading to an immediate degradation of the performance and eventually to disruptions. This aspect is very important and should be properly discussed.

Response:

Thank you for your sincere comments. As discussed earlier, the presence of high-Z components will introduce severe concern in ITER operation due to increased core radiation loss and the easier onset of instabilities. Since this is one of the major challenges in the fusion community, the authors agree that the discussion on this issue is important.

To address this referee's comment, we added a discussion on problems related to metal impurity and potential solutions from this work in the last paragraph of the discussion section (page 9, 2nd paragraph), with the reference suggested by the referee.

Comment 3

The pipeline implemented for the automated 3D-fields optimization, especially concerning the role of machine learning in the optimization. A successful suppression experiment of the ELM crash with a priori modeling with the optimal phase and amplitude of the RMP coil current was already successfully carried out in KSTAR (J.K. Park Nature Physics (2018)). As stated in this paper, “This approach exploits the physics-based optimization scheme of a 3D waveform based on plasma equilibrium and ideal 3D response from GPEC simulation. . . . its computation time, which takes tens of seconds, hinders real-time applicability, limiting its use to pre-programmed or feed-forward strategies.”. As far as I understand from the description in the paper, the machine learning model implemented to overcome such limitations is a surrogate model of the GPEC code (a physics-based model for 3D field optimization). The equilibrium parameters in input to the surrogate model are very likely not changing significantly in what is supposed to be the no-ELM H-mode phase; therefore, they could possibly be calculated beforehand with an inverse equilibrium solver. It appears that the RMP optimizer adaptively changes the current I_{RMP} in real-time by monitoring the ELM state using the D_a signal with a strategy that, as far as the variation of I_{RMP} is concerned, does not seem to require any feedback from the ML component of the algorithm.

Response:

Thank you for raising this question. The equilibrium parameters in a conventional RMP experiment would not change significantly after achieving suppression. However, this is not true for this active RMP control experiment because equilibrium inputs such as β_P , l_i and q_{95} are changing, strongly affecting the 3D field response and coil optimization. This is a universal feature of this I_{RMP} feedback scheme while using hysteresis in the non-ELM phase to improve the confinement. Table 1 shows the sensitivity (how the output relatively changes by input variations) of the ML-3D model to these inputs, illustrating their considerable effect on the optimal coil configuration (or output of ML-3D).

Table 1: Parametric sensitivity of ML-3D output on equilibrium inputs.

Inputs	q_{95}	β_P	l_i	κ	R_X	Z_X
Averaged sensitivity of outputs	0.20	0.20	0.26	0.3	0.09	0.13

Figure 1: Equilibrium parameters of #31873 with integrated ML-3D optimization.

Early RMP phase before ELM suppression: In the early phase (6-8s) of the RMP ramp for getting ELM suppression, β_P significantly decreases due to pedestal degradation. This reduction also influences the plasma current profile, resulting in changes to l_i and q_{95} . As outlined in the manuscript (page 4, 2nd column, line 24), maintaining a small $r_B = B_{\text{core}}/B_{\text{edge}}$ during the early phase is vital for safe access to the ELM-suppressed state. Consequently, real-time optimization considering these varying equilibrium parameters is essential for ensuring successful and safe ELM suppression.

During early ELM suppression: Unlike in conventional RMP experiments, β_P , l_i and q_{95} are evolving due to the feedback control of I_{RMP} . In particular, the feedback control largely enhances confinement (or β_P) during the early phase of ELM suppression. As B_{core} tends to increase with β_P , it becomes important to maintain proper real-time 3D-coil optimization to minimize its adverse effect (by smaller r_B) while simultaneously maximizing confinement. This collaboration between I_{RMP} and the ML-3D control highlights the success of integrated 3D control, facilitating high confinement with safe ELM suppression through field optimization.

Thus, a pre-programmed coil configuration from predictive calculations cannot fully handle a time-varying equilibrium during these two phases. Consequently, a feedback solution of coil configuration from the ML component is required. In addition, the converged state ($>8s$) is the outcome of feedback control, which is also hard to predict before the experiment (unlike the feedforward result), and therefore, the pre-calculation of GPEC for this state is not trivial. The time evolution of β_P , l_i and q_{95} in Figure 5 (manuscript) is also shown in Figure 1, further supporting the importance of ML-3D to deal with changing equilibrium inputs. In addition, it also has the advantage of enabling real-time 3D optimization for unexpected situations, which is essential in long-pulse plasma operations.

To address this referee's comment, we included a discussion on the synergistic benefits of I_{RMP} and ML-3D feedback controls on page 4, 2nd column, line 30 to further clarify and highlight their importance.

Comment 4

The feedback is based on the detection of the ELM, which does not depend on the surrogate model because, if I understand correctly the control scheme and the examples reported, ELM suppression is achieved by a linear increase of RMP linear increase followed by a short plateau once the suppression is achieved to avoid transient oscillations. The degradation should mainly depend on the evolution of the density and temperature pedestals rather than the equilibrium parameters, so my main question is what the surrogate model can add to a pre-computed offline optimization done with the GPEC code coupled to an inverse equilibrium solver, for example. It seems that the main effect is due to the regulation of I_{RMP} while the other parameters, which do not change significantly, need to be optimized for different plasma scenarios. This also seems to be true for the minimization of the 3D fields needed for full suppression of ELMs. Is this correct?

Response:

Thank you again for raising this question. As you mentioned, the primary dynamic of ELM suppression is the pedestal degradation (and increase) by the feedback control of I_{RMP} by ELM detection. Again, pre-computed optimization is available in conventional ELM suppression as an equilibrium parameter doesn't change much just after the ELM suppression. However, the change in the pedestal by feedback control affects the plasma confinement, force balances, and current profiles, eventually leading to a change in equilibrium parameters, including β_P , l_i , q_{95} . Once again, these evolving equilibria necessitate time-varying coil optimization to mitigate the adverse effect of the 3D field while maximizing its efficacy. Therefore, the successful regulation and optimization of I_{RMP} needs the support of optimal coil configuration provided by the surrogate model.

In this manner, high-confinement ELM suppression can be considered as an outcome of integration between I_{RMP} and ML-3D (or GPEC solution) feedback control. Likewise, it also has the advantage of enabling real-time 3D optimization for unexpected situations, which is essential in long-pulse plasma operations. Notably, ML-3D is based on a physics-based model and doesn't require experimental data, enabling its extension to other plasma scenarios and devices without conventional 3D scans and adaptations. This robust applicability to future devices highlights the advantage of the ML-integrated 3D-field optimization scheme.

To address this comment, we added this discussion in page 4, 2nd column, line 30 to further highlight its importance.

Comment 5

Another important question related to this point regards the variability of the scenarios in which the ELMs suppression was tested in the real-time experiments (range of heating power, etc.). I think it would be worth adding more details with respect to this point since this would better qualify aspects such as the robustness and the extrapolability of the developed solution.

Response:

Thank you for your sincere comments. Like other ELM mitigation methods, ELM suppression using RMP has to meet certain operation conditions. Although the precise condition is still not clear, community efforts have identified q_{95} , pedestal density ($n_{e,\text{ped}}$), and heating (which affect collisionality and rotations) as the most important factors. Whenever these conditions are satisfied, ELM control with 3D optimization becomes feasible. Due to the limited opportunity for run time (given the high costs of discharges in DIII-D and KSTAR), we couldn't cover all known ELM suppression windows. Nevertheless, we have managed to test the controller on major windows, covering $q_{95} = 3.3 - 5$, $n_{e,\text{ped}} = 1.5 - 3 \times 10^{19}/m^3$, and beam heating power of 3-6 MW.

To address this referee's comment, we added a brief description of the range of these parameters in the ELM-free database of the Methods section.

Comment 6

Nevertheless, I think that the work described in this paper presents several elements of novelty, it addresses a topic of great interest to the fusion community and not only. It attempts to bridge the gap between theory, simulations, and experiments, taking advantage, among other things, of recent developments in machine learning. The present paper can be considered for publication in this Journal only if the authors adequately address the issues and concerns reported here, including the “minor” points below:

- Why the choice of using H_{89} instead of the most recent $H_{98y,2}$ plasma confinement scaling?
- How have the RMP ramp-up rates and the length of the plateau after ELM suppression been optimized in the two machines? Is there any physics-based reason for their choice?
- Could you elaborate a bit more on the “precursor pattern” in Da and turbulence signals, emerging about 20 ms ahead of ELM reappearances during the suppression phase? Would those turbulence markers be available in real time? Would their processing be feasible in real time if that is the case?

Response:

Thank you for raising these questions.

1) In this study, H_{89} is utilized for the purpose of overlapping (and comparing) the database from both DIII-D and KSTAR tokamaks. Unlike DIII-D, the number of density and temperature profiles in KSTAR is not sufficient to construct an H_{98} database due to limitations in the radial profile diagnostics (which is needed to derive thermal confinement time). Therefore, H_{89} is one of the few effective metrics for confinement quality available in both devices. Still, this choice doesn't change the conclusion drawn from the study, as the H_{98} database in DIII-D (see Figure 2) shows similar results to those presented in Figure 2 (manuscript).

Figure 2: Performance comparison of ELM-free discharges in DIII-D tokamak.

Figure 3: D_α and measured fluctuation (magnetic) during ELM suppression phase

2) The ramp-up rates and the plateau length have not been defined with an exact formula. Instead, we used 1) a time scale greater than the energy confinement time scale (0.05-0.2s in both tokamaks), allowing sufficient time for the kinetic profile to respond to the RMP. Here, we used a longer time scale in KSTAR due to the slower response of the control system imposed by the superconductor magnets. We added a brief description of these choices on page 5, 1st column, last paragraph.

3) The measured turbulence can be observed in temperature, density, and magnetic fluctuations. However, the underlying mechanism of turbulence behavior during RMP

ELM suppression is not fully understood in the fusion community. Figure 3 shows Da shows a dip-like pattern before ELM reappearance. This is a low-frequency feature, and processing is available in real-time.

During the precursor fluctuation signature, broadband fluctuations disappear before the onset of ELM reappearance. Real-time utilization of this pattern is not trivial because it requires real-time digitizers capable of covering high frequencies (up to ~ 300 kHz) and rapid processing capabilities. Recently, new hardware has been installed in both devices (DIII-D and KSTAR) to facilitate high-frequency fluctuation monitoring in real-time, and the ML algorithm for fast signal processing is under development. As this is still ongoing future work, the authors would like to leave this out of the present manuscript. Instead, a brief mention of future work related to fluctuation processing has been added on page 9, 1st column, first paragraph.

Thank you for your support of this manuscript and for the detailed comments that have helped to make the revised text stronger. We hope that with these changes, you can again recommend the manuscript for publication in Nature Communications.

Authors' Response to Reviewer 2

General Comments.

The manuscript (MS) titled "Highest Fusion Performance without Harmful Edge Energy Bursts in Tokamak" describes 3D field optimization, exploiting machine learning and real-time adaptability. The ELM mitigation and the improvement of the figures of merit are vitally important for the fusion reactor in the future, so the MS is of great interest to the plasma society; however, the MS could not be accepted as it is for publication due to the following reasons: It seems the authors confused the concepts of fusion gain and figures of merit. Following the definition $G = \beta_N H_{89} / q_{95}^2$ in the caption of Page 4, I guess that G refers to the figures of merit. The authors should be very careful to re-organize and clarify the fusion gain/figures of merit in the context.

Response:

Thank you for your careful consideration and noting that "*ELM mitigation* and the improvement of the *figures of merit* is vitally important for the fusion reactor in the future". As the figure of merit (G) is a widely used and effective metric for projecting fusion performance (and gain) from current to future devices, including ITER, continuous efforts are underway within the fusion community to enhance its values. We are grateful that the reviewer noticed the error, and we apologize for causing confusion by misuse of fusion gain and G simultaneously. While editing the paper to be readable for the general audience, we made some simplifications, and this incorrect phrase was introduced. We fixed it now by clearly stating G as a figure of merit and removing the expression of fusion gain.

In addition, several of the questions that you raised have led to significant improvements in the clarity of the manuscript contents. We hope that with these modifications and with the detailed responses provided below, you will reconsider your assessment and recommend publication of this work in Nature Communications.

Comment 1

In Fig. 2, a small RMP (DIII-D) region does not overlap with that of non-ELMs (DIII-D). Could the authors explain how they obtained/defined the colored region for RMP DIII-D, KSTAR, and non-ELMs for DIII-D, respectively? And please explain the (possible) reasons. Besides, some points of the Adaptive RMP (DIII-D) are located outside the regions of RMP and non-ELMs for DIII-D. Could the authors provide some explanations? What measures have been taken to achieve a value of G higher than 0.4?

Response:

Thank you for bringing up these questions. In this figure, the RMP (blue) and non-ELM (orange) points represent the different scenarios, where non-ELMs denote operational scenarios without RMP. These colored regions are derived from the data presented in the supplement (see Figure 1 (supplement)). In addition, the blue region only covers the case with conventional RMP, while we have separately indicated the adaptive RMP case with a distinct red marker.

Here, the figure of merit (G) is obtained based on the ITER baseline target parameters ($\beta_N=1.8$, $q_{95}=3$, $H_{89}=2$).

To address this referee's comment,

- 1) We revised the caption of Figure 2 (manuscript) to clarify the categories of data points.
- 2) Added the reason of $G = 0.4$ in the caption of Figure 2 (manuscript).

Comment 2

In Fig. 6 (b) and Fig. 9 (a), the amplitudes of Da are too low, and it is very difficult for the readers to determine the existence of the ELM (during the flattop and decreasing phases of I_{RMP} in Fig.9(a), some mossy ELM seems to exist) or not. Besides, in the paragraph right before the start of II. Discussion, the authors declared that the "increase in $\omega_{tor, ped}$ starts 0.3 s later than the 2.6s that RMP-ramp down starts." is it possible for the authors to exclude the influence of the ELM on the velocity of rotation?

Response:

Thank you for your sincere comments. The scale of the Da figure was set to include the maximum Da peak in the plot. However, the authors do agree with the referee's comment regarding visibility issues. As the referee mentioned, there are some ELMs present in Fig. 6(b) and Fig. 9(a) during the RMP plateau. These mossy ELMs occur irregularly during ELM suppression. Here, they show a much smaller size than Type I ELM and irregular periodicity and disappear in time even with decreasing RMP amplitude, which exhibits different characteristics compared to the ELM-mitigated state. For these reasons, we classified this state during 3D control as ELM-suppressed but with sporadic ELMs.

As the referee mentioned, ELMs can affect the momentum transport and rotation velocity. However, the author thinks that the delayed (0.3s) response of $\omega_{tor, ped}$ shown in Figure 9 (manuscript) is less correlated with ELM dynamics in that 1) ELMs are nearly suppressed before 2.6s, making it difficult for them to give large effect on the momentum transport, 2) no notable rotation change during the I_{RMP} plateau while sporadic ELMs keep decreasing, 3) last sporadic ELM occurs 0.3s prior to the $\omega_{tor, ped}$ response, which is longer than the typical momentum confinement time scale ($\sim 0.1-0.2$ s), and 4) no notable change in ELM properties during 2.6-3.0 s. **To address this referee's comment,**

1) We have re-scaled the y-axis of the plots, making the Da signal three times larger to enhance visibility.

2) We have mentioned a weak correlation between rotation response and ELM dynamics in page 8, 1st column, last paragraph to improve the clarity.

Comment 3

In Fig. 7, the maximum enhancement of KSTAR is around 50% whereas $\sim 90\%$ for DIII-D. Could the authors clarify the reason?

Response:

Thank you for raising this question. As outlined in the manuscript, the main focus of this paper is the new control approach for ELM mitigation with optimized fusion confinement (and G), and how far the control can be achieved while its physics understanding is also important. We know the process of increase in confinement. However, for the exact level of increase, although we have a case for a physics explanation, we do not think we have a high confidence level to support that claim. This is due to 1) lack (and uncertainties) of diagnostics (profile and flow) and 2) complex physics of ELM suppression, allowing possible multiple explanations. With this caution, below, we state the most plausible explanation.

Throughout the optimization process by feedback 3D control, combined nonlinear physics dynamics, such as 3D response, profile variations, and flow evolutions, come into play, affecting the strength of RMP-hysteresis and confinement enhancement. Despite the difficulties mentioned above, we posit that the notable confinement recovery observed in DIII-D is a result of the favorable flow effect from the $\omega_{E,10/3} \rightarrow 0$ during $\omega_{\text{tor, ped}}$ recovery, as discussed in the paragraphs on page 7 and 8. In this regard, we speculate the difference between DIII-D and KSTAR also stems from the flow effect, where ω_E profile in KSTAR is not favorable in leveraging the $\omega_{\text{tor, ped}}$ recovery. Unfortunately, KSTAR lacks a stable flow diagnostic for ω_E and, therefore, cannot compare ω_E profiles with DIII-D cases.

Therefore, a conclusive understanding of this physics mechanism will possibly be answered with future work, including diagnostic upgrades and delicate profile datasets. In this manner, the authors would like to leave this out of the present manuscript and highlight the importance of this future work, as mentioned in page 8, last paragraph before the discussion section.

Figure 4: Zoomed plot of Figure 6 (manuscript)

Comment 4

In Fig. 6 (d) (e), the ion/electron temperature drops when the ELM occurs; meanwhile, the electron density rises abruptly; one might draw that the pressure might be constant during the ELM burst. However, this phenomenon could not be observed on KSTAR (Fig. 6 (i) and (j)). Could the authors clarify the phenomenon?

Response:

Thank you for addressing this question. As noted by the referee, in Fig. 6 (d) (e), the temperature drops (and density increases) after the first ELM crash, as shown in Figure 4. However, temperature (and density) in Fig. 6 (i) (j) drops during I_{RMP} ramp, which is far after the first ELM crash. Here, these two cases (#190736 vs. #26004) exhibit two major differences in their operation conditions: 1) q_{95} 3.3 vs. 5 and 2) energy confinement time ~ 100 vs. 50ms.

The temperature drops in #190736 can be considered as an outcome of a rapid increase in $n_{e,\text{eped}}$, as the temperature pedestal can drop to meet the temporal power balance (like an adiabatic response). This is also consistent with the gradual evolution of pedestal pressure. The rapid recovery of $n_{e,\text{eped}}$ can be understood as combined results of 1) ELMs crash, which flush out the RMP-induced transport (e.g., island, turbulence, and others) at the pedestal, allowing faster $n_{e,\text{epd}}$ recovery, and 2) transient sawteeth followed by ELM oscillation, making plasma evolution more rapid (or transient). Here, sawteeth is easier to occur in this case because of lower q_{95} . #26004 also shows the faster recovery of $n_{e,\text{epd}}$ after the ELM crash, but it is still a gradual change due to the absence of sawteeth, and therefore, a drop in temperature doesn't occur.

On the other hand, the fast drop of temperature and density pedestal in #26004 can be considered as a consequence of the re-established strong RMP-induced transport during I_{RMP} ramp ($>7.8\text{s}$), leading to the recovery of ELM suppression. Although this trend in #190736 is a bit blurred by rapid density increase after the first ELM crash, it also exhibits a decrease in temperature and density pedestal during I_{RMP} ramp ($>3.1\text{s}$), as shown in Fig. 6 (d) (e). Here, #26004 shows a much faster decrease due to a shorter confinement time scale, allowing quicker profile evolution. Therefore, considering these aspects, the observed differences in the two cases are likely due to different plasma parameters and core transient events (sawteeth) rather than general differences in the device. **To address this referee's comment**, we briefly mentioned the rapid density increase in #190736 on last paragraph in page 5.

Comment 5

Typos. Caption of Fig.5, c H89(blue)..... should be b H89(blue)

Caption of Fig. 8. q_95 should be q_{95}

Caption of Fig.9. c $T_{e,ped}$ should be d $T_{e,ped}$

Response:

Thank you for your sincere comments. These typos have been corrected.

Thank you for your detailed comments on this manuscript. We have worked through all of them and made improvements to the text and figures. With these modifications, we hope you will reconsider the suitability of the manuscript for Nature Communications.

Authors' Response to Reviewer 3

General Comments. Introduction, key results, and significance One of the main operational hurdles for a tokamak concept of magnetic confinement fusion reactor is the avoidance of high transient heat fluxes towards the plasma-facing components that are the consequence of edge localized mode (ELM) instabilities. Therefore, the fusion research community worldwide has dedicated a considerable effort to understanding the physics behind these phenomena and developing methods to mitigate the associated heat fluxes to acceptable levels for the long-term operation of a future fusion reactor. ELM control by resonant magnetic perturbation (RMP) field is one of the first methods to have provided satisfactory results in that regard and was thus chosen as the main ELM control scheme for tokamak ITER. However, the ELM suppression by RMP comes at the cost of the reduced pressure of the confined plasma as a consequence of the increase in plasma transport caused by the stochastization of the magnetic field lines in the edge plasma. Furthermore, the full ELM suppression is typically accessible only in a narrow plasma and RMP parameter space. Thus, other methods for ELM heat flux control have recently been considered and investigated. These include but are not limited to divertor detachment or utilization of plasma confinement regimes where ELMs are absent altogether.

General Comments. In this manuscript, the authors present a solution to significantly limit the RMP ELM control drawbacks mentioned above. They do so by utilizing a linear description of edge vs. core coupling of RMP to plasma, based on an ideal magnetohydrodynamic approximation that has been recently reported to achieve very successful predictive results for ELM suppression on KSTAR tokamak. Within this manuscript, the authors also demonstrate this method to work on the tokamak DIII-D. The novelty of the manuscript also resides in the authors having utilized the machine learning approach to design a control algorithm to regulate the RMP field spatial distribution and magnitude - to maintain the ELM suppression during the changing conditions of tokamak discharge and to limit the necessary perturbation of plasma. Up till this point this has only been done by setting a predefined RMP current waveforms and spatial distribution before the discharge took place, hence this automation presents an important step ahead in the ELM control. Furthermore, it is shown in the manuscript that limiting the RMP-induced plasma confinement deterioration by this method results in the level of plasma confinement that is highly competitive, if not superior, to the alternative methods for ELM control. By doing so, the authors challenge the paradigm among many members of the fusion research community that ELM suppression by RMP is difficult to be reliably guaranteed in experimental practice and that the associated deterioration of the H-mode plasma confinement would be too much of a tradeoff to be viable for a reactor.

I thus recommend that the manuscript be published pending minor revisions. Among those, my main comment relates to the limitation of the chosen methodology for ELM detection when extrapolated to a fusion reactor (see Data and methodology).

Response:

Thank you for your overall endorsement of this manuscript and for your close attention to this manuscript and its relationship to previous work on related subjects. Your comments

have helped to strengthen the manuscript by clarifying and broadening the message of the article.

Comment 1

Validity and robustness of data

The data presented by the authors appears trustworthy. Tokamaks are complex, large physical experiments operated by dozens of physicists and engineers. Although the full details of such experimental arrangement and plasma therein are challenging to encompass within a single publication, the presence or absence of ELMs is very evident within the experiment operation practice. Fig. 9, in particular, is very convincing of the validity of the method and correct interpretation of the data as it correctly shows well-known degradation of plasma density due to RMP pulse improvement in plasma temperature (a new consequence of the optimization method presented by the authors) and, most importantly, correct distinction of the ELM-suppressed regime from transient ELM-free phase where plasma density ramps up (that otherwise could have been argued to be the cause of the confinement improvement).

Response:

Thank you for your sincere comment. We tried to avoid any potentially misleading representations in the data analysis and are glad that this effort was successful.

Comment 2

Data and methodology

To detect the presence or reappearance of the ELMs for the real-time optimization algorithm to adjust the RMP accordingly, the authors rely on the bursts of the Da radiation signal. This signal originates from the interaction of hot plasma particles released from a confined region during the ELM event and the hydrogen (isotopes) deposited on the first wall of the tokamak (typically the divertor). Such a method is indeed viable for the majority of the present-day tokamaks (including KSTAR and DIII-D – the two devices utilized in the manuscript) since these typically use graphite as a material for the first wall (see, e.g. [1]) that is known for high long-term retention of hydrogen. However, the authors aim to provide a robust RMP ELM suppression control scheme for future reactors ITER, DEMO, and possibly beyond. ITER is presently investigating the consequences of switching from a beryllium first wall to tungsten – see the press release by the Head of the Science Division of ITER from October 2023 [2] and numerous presentations and posters by the ITER Organization at recent plasma physics conferences. One of the reasons behind this decision is tungsten has a lower retention of hydrogen (tritium, in particular) – see review [3] and references therein. For this reason, the Da signal on ITER will not be as reliable for ELM detection as on present-day KSTAR and DIII-D tokamaks. An example of this can be seen at tokamak ASDEX Upgrade – a device operating with a tungsten-coated first wall since 2007 [4], particularly in their first report of ELM suppression by RMP [5]. Thus, in the more recent ELM suppression experiments, the ASDEX Upgrade team has relied on the divertor thermoelectric currents to detect the presence and absence of ELMs [6]. The authors should thus comment in the manuscript on the viability of the Da signal as an ELM indicator in tungsten first wall tokamaks and on the possibility of expanding the controller of their algorithm for other, more reactor-relevant ELM detection methods.

Response:

Thank you for your sincere comment. The authors agree obtaining a Da signal may be more difficult in ITER and future devices with high-Z metallic (like tungsten) walls. To address this issue, the algorithm can utilize alternative signals, such as divertor thermo-electric currents (which the referee mentioned) and also fast temperature diagnostics (electro-cyclotron emission or interferometry), which can capture the short perturbation by ELMs. In particular, these diagnostics could be favorable for 3D optimization in that this can distinguish the ELMs induced by core instability (or perturbation) and 3D control. For example, as shown in Figure 8 (manuscript), sporadic ELMs by sawtooth deteriorate the 3D optimization performance. Here, if the algorithm can differentiate between ELMs induced by control feedback (such as those caused by I_{RMP} feedback control) and those resulting from sawtooth events, the control efficiency can increase regardless of sawteeth, resulting in higher plasma performance.

To address this referee's comment, we added the brief discussion on diagnostics on page 7, 1st column, line 35.

Comment 3

Furthermore, the authors state in the Methods section about the ideal plasma response calculation by GPEC that the core response part extends to $\psi_N = 0.9$ all the way from $\psi_N = 0.0$. This is not generally true since the calculation of the core coupling part by GPEC starts outside the 1/1 resonant if it is present - not at the magnetic axis of $\psi_N = 0.0$.

Response:

Thank you for your sincere comment. As the referee mentioned, in practice, the starting range is not always $\psi_N = 0.0$ as we remove the $q = 1/1$ not to include the 1/1 resonant

component. Here, we stated $\psi_N = 0-0.9$ for simplicity as 1/1 is automatically removed during the GPEC simulation.

To address this referee's comment, we added a description of removing the 1/1 component in the Ideal plasma calculation in the Method section.

Comment 4

Clarity and context

To a general reader, the authors provide a very good introduction to the topic of plasma confinement, ELMs, and their control by RMP. However, the introduction is missing a brief discussion of other alternatives for resolving the ELM heat fluxes and their performance in comparison to the method proposed by authors - although this is discussed later, on page 6. The impact of the manuscript on its respective field would be better conveyed to a more specialized reader if either Fig. 2 in the manuscript was replaced by Fig. 1 from the supplement or if the above-mentioned conclusions from page 6 were also stated in the introduction.

Response:

Thank you for your sincere suggestion. The authors agree that Figure 1 (supplement) will deliver better information to the readers in the fusion community. Still, the author would like to keep Figure 2 for the general reader as it has easier visibility.

Instead, to address this referee's comment, we added the description of small and non-ELMing regimes in the 2nd paragraph of Introduction section to help readers get a better understanding of the figure.

Comment 5

Other than that, I also have minor comments regarding the clarity of some statements in the manuscript:

- First paragraph, page 2: “produce” is confusing and should be replaced by e.g. “achieve”, as the related term “fusion triple product” does not refer to production, but to product as a result of multiplication.
- Second paragraph, page 2: It is not clear which particular “significant advantage” the authors refer to when mentioning the stabilization of the ELMs by RMP on ITER. Needs more details.
- Section I. Results, first paragraph: The relation between high B_{core} and the plasma disruption is not known to a general reader not familiar with RMP experiments on tokamaks. The authors need to elaborate more.

Response:

Thank you for your sincere suggestion.

To address this referee’s comment,

- 1) We have replaced “produce” with “achieve”.
- 2) We have added further detail of advantages in that the RMP scheme is an ‘active’ solution for mitigating the ELMy heat flux and ELM-induced seed perturbation, which can drive core instability. This description is added on page 2, 2nd column, first paragraph.
- 3) We have added a description that “ B_{core} induces seed perturbations and radial transport in the core, making the onset of plasma disruption easier” in the corresponding paragraph.

Comment 6

- Same paragraph: Single disruption terminating the life of a fusion reactor is too strong of a statement. Although it is true that the number of disruptions will have to be significantly limited in a reactor, it is likely (taking ITER as the closest example) that a disruption budget will be formulated for a reactor detailing how many disruptions of which plasma scenarios can the device withstand - see, e.g. [7]. Furthermore, disruptions due to high Bcore are typically preceded by a detectable precursor signature [8] that allows disruption mitigation mechanisms to be engaged to ensure a smoother termination of the discharge and to limit the extent of possible damage to the machine [9].
- Page 4: Phrase “totally new... path” should be replaced by a more neutral form, e.g., “alternative... path”. Likewise, in the next sentence it would be better to refer to the plot, to replace “It is clear” with e.g. “from the plot it is apparent”.
- Page 6: the general sentence “few ELMs are tolerable in future fusion machines” should be supported by a specific reference to, e.g., the ITER budget for allowed ELM size and count.
- Discussion section, first paragraph: in “achieving the highest fusion among various ELM-free scenarios...” I assume the authors intended to write “fusion product”?

Response:

Thank you for your sincere comments.

To address this referee’s comment,

- 1) We have reduced the strength of the statement to “an unmitigated high current disruption can greatly reduce the machine lifespan.”, adding the suggested reference.
- 2) We have changed the expressions “effective” and “From the plot, it is apparent”
- 3) Based on Ref.[60], the ITER budget roughly allows a few ELMs (2-8, depending on the scenarios). We added this reference to the manuscript.
- 4) We have corrected it to G .

Thank you for your support of this manuscript and for the detailed comments that have helped to make the revised text stronger. We hope that with these changes, you can again recommend the manuscript for publication in Nature Communications.

Reviewer #1 (Remarks to the Author):

Most of the concerns raised by the reviewers have been addressed by the authors in this new version of the manuscript, and I appreciate the discussion included in the final discussion section on problems related to metal impurity accumulation. As correctly stated by the authors, ELM suppression using RMP has to meet certain operating conditions, and there are many uncertainties on the applicability of the proposed approach on future devices.

However, it remains unclear for the general reader how effectively the proposed strategy based on RMP for ELM-free scenarios can be applied in future devices and in reactor-relevant conditions (among things, reprising other reviewers' comments, the improvement in terms of confinement and plasma performance in DIII-D - a non-metallic device - and KSTAR are quite different). I suggest the authors to discuss in a more explicit and structured way operating conditions, range of applicability and limits of the proposed approach. The integrations made in the paper don't yet allow a non-expert fusion reader to have a clear idea of the overall context.

Figure 7 should be redrawn for better readability: the pulse numbers are too small and overlapping, and it is basically impossible to see the $n=2$ triangle for KSTAR.

I therefore recommend the revised manuscript be considered for publication only if the above concerns are properly addressed.

Reviewer #2 (Remarks to the Author):

Thank you for the extensive revision work. I have checked the revised version and found its suitability for publication. Please update the following two small flaws:

- 1, Page 2, "The predicted heat energy reach....." shall be "The predicted heat energy reaches.....";
- 2, In the caption of Fig.8, q_{95} shall be q_{95} .

Reviewer #3 (Remarks to the Author):

Having read the new, updated manuscript and the associated document with the response to the reviewers in detail, I can confirm that my past comments have been addressed appropriately - by directly addressing the topic in a sufficient manner, or using a proper reference to other work. I have no new comments regarding the revised form of the document.

Although in the response "2)" to the "Comment 6" the authors made a typo in the manuscript by writing "effecitve" instead of "effective", it is something that can be easily corrected.

I thus recommend the revised manuscript for publication.

Responses to Reviewers' Comments for Manuscript NCOMMS-23-50991A

Highest Fusion Performance without Harmful Edge Energy Bursts in Tokamak

Addressed Comments for Publication to

by

SangKyeun Kim, et al.

Authors' Response to Reviewer 1

General Comments. Most of the concerns raised by the reviewers have been addressed by the authors in this new version of the manuscript, and I appreciate the discussion included in the final discussion section on problems related to metal impurity accumulation. As correctly stated by the authors, ELM suppression using RMP has to meet certain operating conditions, and there are many uncertainties on the applicability of the proposed approach on future devices.

Response:

We thank the referee for the detailed feedback you gave to the text of this manuscript. We tried our best to answer the raised comments. Your thoughtful comments are greatly appreciated and have led to the improvement of the manuscript's contents.

Comment 1

However, it remains unclear for the general reader how effectively the proposed strategy based on RMP for ELM-free scenarios can be applied in future devices and in reactor-relevant conditions (among things, reprising other reviewers' comments, the improvement in terms of confinement and plasma performance in DIII-D - a non-metallic device - and KSTAR are quite different).

Response:

We thank the referee for their comment. As you mentioned, two issues can be considered regarding the application of this strategy on future devices: Control scheme and Scenario accessibility.

As described in the manuscript, we tested the controller in two tokamaks (KSTAR and DIII-D are non-metallic carbon devices) with various RMP-ELM suppression regimes,

including $n = 1, 2, 3$ of RMP, and found that the controller is compatible in all cases. While this approach still has limitations to be improved, which are described in the Discussion section, these successful demonstrations show that the proposed strategy is effective when the plasma meets the conditions for accessing an ELM-free (or suppressed) scenario by RMPs.

The issue of accessibility of the ELM suppression state is more closely related to a physics discussion that may be beyond the scope of this paper, which focuses on control methods and performance. Here, previous studies have shown that this accessibility requires specific plasma conditions, which can impose constraints on reactor design and introduce uncertainty about whether these conditions can be achieved. Here, a recent modeling study [1] suggests that these conditions are attainable in ITER, identifying the RMP as one of the leading approaches for ELM control. Consistently, large ongoing projects such as ITER ($Q > 10$, [2]) and SPARC ($Q > 5$, [3,4]) seem not to be largely affected by this constraint. They are planning with RMP as the primary method based on extensive physical analysis and are designing their devices to enable scenarios without ELM based on RMP. Therefore, if scenario accessibility is achieved through this design effort, then the proposed control strategy will also be compatible. In addition, future advances in theory and modeling will provide further light on this issue.

To address this referee's comment, we added this discussion in the second paragraph of the Discussion section to clearly convey these ideas to non-expert readers.

Comment 2

I suggest the authors to discuss in a more explicit and structured way operating conditions, range of applicability and limits of the proposed approach. The integrations made in the paper don't yet allow a non-expert fusion reader to have a clear idea of the overall context.

Response:

We thank the referee for their comment. As you mentioned, the discussion in the former version about the limitations of this work was insufficient to convey a clear idea to readers who are non-fusion experts.

As described in the manuscript, the proposed control strategy has proven to be promising but still has challenges and uncertainties, including

- 1) Uncertainty of robust access to ELM suppression with RMP in future devices.
- 2) Question on the efficiency of the proposed method in metallic devices. Here, DIII-D and KSTAR (the wall is partially upgraded to tungsten this year) are both carbon devices, and it is true that metallic devices may affect the control results. This will be answered by extending the application of this control method to existing and upcoming metallic tokamaks.
- 3) Possible difficulty for fully adaptive RMP optimization in future devices.

To address this referee's comment, we addressed the above **issues and potential solutions** in each paragraph in the **Discussion section**. In addition, we've reorganized the **Discussion section** and added a **Last paragraph** to improve readability for non-expert readers and to clearly convey these issues and potential solutions.

Comment 3

Figure 7 should be redrawn for better readability: the pulse numbers are too small and overlapping, and it is basically impossible to see the $n=2$ triangle for KSTAR. I therefore recommend the revised manuscript be considered for publication only if the above concerns are properly addressed.

Response:

We thank the referee for their comment. As you mentioned, the visibility of Figure 7 is insufficient for the readers due to the overlap of data and labels.

To address this referee's comment, we renewed the figure by

- 1) Increasing the text size and improving data selection and distribution in the plot to minimize the overlap of scattered points and labels while maintaining the conclusion of the figure.
- 2) Changing the scale of the X and Y axis to the log scale for better visibility.

We thank the referee for further comments on this manuscript. We have worked to clarify the applicability and limits of work and improve the figure. We hope that with these changes, you can again recommend the manuscript for publication in Nature Communications.

Authors' Response to Reviewer 2

General Comments.

Thank you for the extensive revision work. I have checked the revised version and found its suitability for publication. Please update the following two small flaws:

1, Page 2, "The predicted heat energy reach....." shall be "The predicted heat energy reaches.....";

2, In the caption of Fig.8, q_{95} shall be q_{95} .

Response:

We deeply appreciate the reviewer's past comments, which have greatly contributed to the advancement of our paper. We included these new comments in the manuscript. We hope that with these changes, you can once again consider recommending the manuscript for publication in Nature Communications.

Authors' Response to Reviewer 3

General Comments. Having read the new, updated manuscript and the associated document with the response to the reviewers in detail, I can confirm that my past comments have been addressed appropriately - by directly addressing the topic in a sufficient manner or using a proper reference to other work. I have no new comments regarding the revised form of the document. Although in the response "2)" to the "Comment 6" the authors made a typo in the manuscript by writing "effecitve" instead of "effective", it is something that can be easily corrected. I thus recommend the revised manuscript for publication.

Response:

We are thankful for the reviewer's previous comments, which have significantly enhanced our manuscript. We included the new comment in the manuscript. We hope that with these changes, you can again recommend the manuscript for publication in Nature Communications.